# Model Collapse Demystified: The Case of Regression

**Elvis Dohmatob**[§][*]    **Yunzhen Feng**[†]    **Julia Kempe**[§][†][‡]
[§] FAIR, Meta
[†] Center for Data Science, New York University
[‡] Courant Institue of Mathematical Sciences, New York University
[*] Correspondence to dohmatob@meta.com

## Abstract

The era of proliferation of large language and image generation models begs the question of what happens if models are trained on the synthesized outputs of other models. The phenomenon of "model collapse" refers to the situation whereby as a model is trained recursively on data generated from previous generations of itself over time, its performance degrades until the model eventually becomes completely useless, i.e. the model collapses. In this work, we investigate this phenomenon within the context of high-dimensional regression with Gaussian data, considering both low- and high-dimensional asymptotics. We derive analytical formulas that quantitatively describe this phenomenon in both under-parameterized and over-parameterized regimes. We show how test error increases linearly in the number of model iterations in terms of all problem hyperparameters (covariance spectrum, regularization, label noise level, dataset size) and further isolate how model collapse affects both bias and variance terms in our setup. We show that even in the noise-free case, catastrophic (exponentially fast) model-collapse can happen in the over-parametrized regime. In the special case of polynomial decaying spectral and source conditions, we obtain modified scaling laws which exhibit new crossover phenomena from fast to slow rates. We also propose a simple strategy based on adaptive regularization to mitigate model collapse. Our theoretical results are validated with experiments.

## 1 Introduction

*Model collapse* describes the situation where the performance of large language models (LLMs) or large image generators degrade as more and more AI-generated data becomes present in their training dataset [44]. Indeed, in the early stages of the generative AI evolution (e.g the ChatGPT-xyz series of models), there is emerging evidence suggesting that retraining a generative AI model on its own outputs can lead to various anomalies in the model's later outputs. This phenomenon has been particularly observed in LLMs, where retraining on their generated content introduces irreparable defects, resulting in what is known as "model collapse", the production of nonsensical or gibberish output [44, 8]. Though several recent works demonstrate facets of this phenomenon *empirically* in various settings [23, 32, 33, 8, 9, 21], a theoretical understanding is still missing.

In this work, we initiate a theoretical study of model collapse in the setting of high-dimensional supervised-learning with linear regression. This is equivalent to kernel regression[1], which serves as an effective proxy for neural networks in various regimes, for instance in the infinite-width limit [37, 49, 25, 28] or in the lazy regime of training [12]. [11] characterize the power-law generalization error of regularized least-squares kernel algorithms, assuming a power-decay spectrum of the kernel (capacity) and of the coefficients of the target function (source). Source and capacity power decay

---

[1] we present the linear regression setting for ease, the extension to the kernel setting is straightforward.

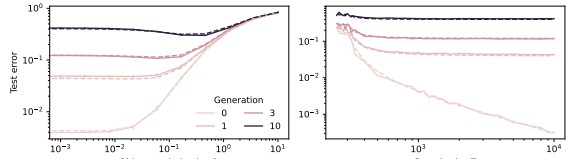
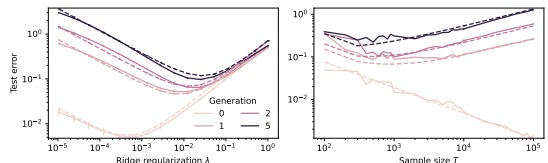

(a) **Isotropic covariance spectrum** $\Sigma = I_d$**.** Here, we show the evolution of test error for different sample size ($T$), different levels of ridge-regularization ($\lambda$), and training data from different generations ($n$) of fake data. The setup is: input-dimension $d = 300$, sample size for fake data generator $T_0 = 600$, noise levels $\sigma = 0.1$ and $\sigma_0 = 0.2$. **Left plot** is for $T = 1000$ and different values of $\lambda$. Notice the U-shape of the curves for large values of $n$, indicating the existence of a sweet spot (optimal regularization parameter). **Right plot** is for $\lambda = 10^{-3}$ and different values of $T$. The broken lines correspond to the theoretical result established in Theorem 4.1.

(b) **Power-law covariance spectrum.** Refer to Eqn (23). The setup is: $d = 300, T_0 = 600, \sigma = \sigma_0 = 1$, $\Sigma = \text{diag}(\lambda_1, \ldots, \lambda_d)$, where $\lambda_k \propto k^{-2}$. **Left plot** corresponds to $T = 10,000$ and **Right plot** corresponds to adaptive regularization $\lambda = T^{-\ell_{crit}}$, where $\lambda = \lambda(T)$ as proposed in [14]. See Section D for details. The broken curves are as predicted by our Theorem 5.1. Though $\ell = \ell_{crit}$ is optimal in classical case, it is not in the setup of model collapse. In fact here, the test error diverges with sample size $T$. Our theory proposes a corrected value of this exponent which gracefully adapts to synthesized data. See Figure 4 (Appendix) for results on MNIST [16].

Figure 1: **Demystifying model collapse.** Refer to Appendix D for details on the experimental setup.

capture properties of the data and the model that give rise to power law scaling of test error in terms of data set size and model capacity, as empirically observed e.g. in [26, 24]. More recently, scaling laws have been shown for kernel models under the Gaussian design, e.g. in [46, 13, 14] for regression and [15] for classification. [39, 43, 31] study scaling laws for regression in the random feature model.

**Summary of Main Contributions.** Following the rich tradition in prior works outlined above, we study the Gaussian design where the input $x$ is sampled from a multivariate zero-mean Gaussian $\mathcal{N}(0, \Sigma)$ and labels $y$ are determined by a linear ground truth function with independent label noise $\epsilon$ as $y = x^\top w_0 + \epsilon$ (we present the linear regression setting for ease, the generalization to the kernel setting is straightforward). At each generation step, an approximation to $w_0$ is learned from the data, and used to generate new, fake /synthetic labels for the next generation. Note that the machine learner has no control over the fake data generation process. It only sees data from a stage $n$ of this process, which is then used to fit a downstream predictor. Our main findings can be summarized as follows:

*(1) Exact Characterization of Test Error under Iterative Retraining on Synthesized Data.* In Section 4 (Theorem 4.3), we obtain analytic formulae for test error under the influence of training data with fake / synthesized labels. For $n$-fold iteration of data-generation, this formula writes

$$E_{test} = E_{test}^{clean} + \Delta Bias + n \cdot \sigma_0 \rho(\lambda, T, T_0, \sigma, \Sigma), \tag{1}$$

where $E_{test}^{clean}$ is the usual test error of the model trained on clean data (not AI-generated) and $\sigma_0^2$ the label noise level in the clean data distribution. The non-negative term $\rho$ precisely highlights the effects of all the relevant problems parameters: the feature covariance matrix $\Sigma$, sample size $T$, original data size $T_0$, label noise level in the fake data distribution $\sigma^2$, and regularization $\lambda$. The non-negative term $\Delta Bias$ is an increase in bias brought about by the iterative synthetic data generation process. This term disappears in the *under-parametrized* regime (Corollary 4.4), if each stage in the process was fitted on sufficiently many samples $T_0$ compared to the input dimension $d$ (i.e if $T_0 \geq d$). In the *over-parametrized* case where $T_0 < d$, this term is either a constant (Theorem 4.5) or an increasing function of $n$, depending on whether the design matrix stays the same or is resampled across different generations (Theorem 4.6). Notably, even in the case of noiseless labels (when $\sigma_0 = 0$), the downstream model converges to a Gaussian process around zero exponentially fast with the number of iterations $n$, leading to "catastrophic" model collapse.

A direct consequence of (1) is that, as the number of generations $n$ becomes large, the effect of re-synthesizing will make learning impossible. We note that the multiplicative degradation in scaling with the number of generations $n$ is completely analogous to what has been shown in [18] for infinite memory models and their variants and empirically observed there. Illustration in Figures 1a and 2.

*(2) Modified Scaling Laws.* Turning to the special case of power-law spectra of the covariance matrix $\Sigma$, which allows to derive test-error scaling laws [11, 46, 14, 29], we obtain in Section 5 (see Theorem 5.1) precise new scaling laws of the test error that quantitatively highlight the negative effect of training

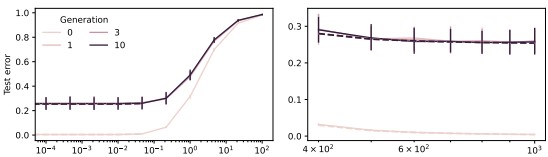 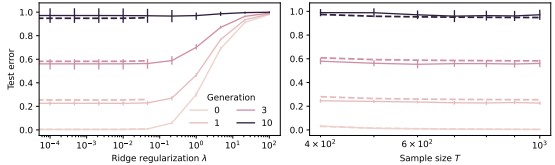

(a) Identical intermediate design matrices $X_n = X_0 \, \forall n$.  (b) Independent intermediate design matrices.

Figure 2: **Model collapse in the case of noiseless over-parametrized synthetic data generator.** Here $d = 300$, the sample sizes for the different versions of the fake data generator are equal, i.e $T_n = T_0 = d/2$ for all $n$, and noise levels are $\sigma_0 = 0$ and $\sigma = 0.1$. Everything else is as in the setting of Figure 1a. Broken lines correspond to the theoretical estimates given in Theorem 4.3. As predicted by our theory, the test error of the model fitted on synthetic data ($n \geq 1$) increases (relative to the baseline $n = 0$, corresponding to training on clean data). The model collapse here, even in the absence of noise ($\sigma_0 = 0$), is due to the fact that the synthetic data-generator does not have access to enough data to capture the true labelling function. **(a)** Importantly, and in accordance to our theory, the amount of model collapse in the case $X_n \equiv X_0$ is due to an increase in bias term of the test error of the model and does not depend on the number of generations $n$ as long as $n \geq 1$. **(b)** In contrast, for the case where the $X_n$'s are independent, the increase in bias term grows with $n$, leading to "catastrophic" model collapse (Theorem 4.6). Refer to Appendix D for the experimental setup.

on synthetically generated data. Further exploiting our analytic estimates, we obtain (Corollary 5.2) the optimal ridge regularization parameter $\lambda$ as a function of all the problem parameters (sample size, spectral exponents, strength of fake data-generator, etc.). This new regularization parameter corresponds to a correction of the the value proposed in the classical theory on clean data [14], and highlights a novel crossover phenomenon where for an appropriate tuning of the regularization parameter, the effect of training on fake data is a degradation of the fast error rate in the noiseless regime [14, 11] to a much slower error rate which depends on the amount of true data on which the fake data-generator was trained in the first place. On the other hand, a choice of regularization which is optimal for the classical setting (training on real data), might lead to catastrophic failure: the test error diverges. See Figure 1b for an illustration.

## 2 Related Work

Current LLMs [17, 30, 10, 47], including GPT-4 [1], were trained on predominantly human-generated text; similarly, diffusion models like DALL-E [40], Stable Diffusion [42], Midjourney [35] are trained on web-scale image datasets. Their training corpora already potentially exhaust all the available clean data on the internet. A growing number of synthetic data generated with these increasingly popular models starts to populate the web, often indistinguishable from "real" data. Recent works call attention to the potential dramatic deterioration in the resulting models, an effect referred to as *"model collapse"* [44].

Empirical evidence of model collapse has been reported across various domains [23, 32, 33, 8, 9, 21]. Some theoretical studies [44, 7, 2, 18] have begun exploring this phenomenon. [44] attribute collapse to finite sampling bias and function approximation errors in the (single) Gaussian case but only provide lower bounds without detailed analytic expressions. [7] analyze the training process at the distribution level using both clean and synthetic data and provide stability results. However, these results do not account for finite samples and are only valid locally in parameter space, making them more relevant to fine-tuning rather than training from scratch. [2] examine "self-consuming loops" in the Gaussian case by assuming a sampling bias that reduces data variance with each generation—a (martingale) assumption that we do not require. These studies lack a comprehensive theoretical framework to quantify model collapse and its impact on scaling laws. Our work addresses these gaps by providing an analytic theory that captures how model collapse emerges from training on synthetic data, providing a deeper understanding that goes beyond merely identifying the collapse. A concurrent study by [18] demonstrate that model collapse in foundation models can be attributed to a breakdown in scaling laws [26, 24], where increasing the sample size eventually fails to improve model performance. This finding, theoretically shown for discrete data in variants of the infinite memory model, complements our analytical results on how synthetic data alters the rate of scaling laws, as discussed in Section 5.

## 3 Theoretical Setup

We now present a setup which is simple enough to be analytically tractable, but rich enough to exhibit a wide range of regimes to illustrate a range of new phenomena that emerge with model collapse.

**Data Distribution and Synthetized Data.** Consider the distribution $P_{\Sigma, w_0, \sigma^2}$ on $\mathbb{R}^d \times \mathbb{R}$ given by

$$\textbf{(Input)} \ x \sim N(0, \Sigma), \quad \textbf{(Noise)} \ \epsilon \sim N(0, \sigma^2), \ \text{indep. of } x \quad \textbf{(Output/Label)} \ y = x^\top w_0 + \epsilon. \tag{2}$$

The positive integer $d$ is the input-dimension, the vector $w_0 \in \mathbb{R}^d$ defines the ground-truth labelling function $x \mapsto x^\top w_0$, the matrix $\Sigma \in \mathbb{R}^{d \times d}$ is the covariance structure of the inputs. The scalar $\sigma^2$ is the level of label noise. Here, we consider the linear case for clarity. We describe the extension to the kernel setting in Appendix C. Thus, in classical linear regression, given a sample $(X, Y) \equiv \{(x_1, y_1), \ldots, (x_T, y_T)\}$ of size $T$ from $P_{\Sigma, w_0, \sigma^2}$, one seeks a linear model $\widehat{w} \in \mathbb{R}^d$ with small test error

$$E_{test}(\widehat{w}) := \mathbb{E}_{x,y}[(x^\top \widehat{w} - y)^2] - \sigma^2 = \|\widehat{w} - w_0\|_\Sigma^2, \tag{3}$$

where $(x, y) \sim P_{\Sigma, w_0, \sigma^2}$ is a random clean test point. In our setup for studying model collapse, the training data $(X, Y)$ is sampled from an iterative loop where each generation of the model serves as the labeller for the data for the next generation. This process is described below.

---

**Structure of the Synthesized / Fake Data Generator.** Consider a sequence of data distributions

$$P_{\Sigma, w_0, \sigma_0^2} \to P_{\Sigma, \widehat{w}_1, \sigma_1^2} \to \ldots \to P_{\Sigma, \widehat{w}_n, \sigma_n^2 \to \ldots}, \tag{4}$$

where $\widehat{w}_n$'s is defined recursively by $\widehat{w}_n = w_0$, and

$$\widehat{w}_n = \text{Fit}(X_{n-1}, \overline{Y}_{n-1}), \ \text{for } n \geq 1, \tag{5}$$

where $\overline{Y}_n := X_n \widehat{w}_n + E_n$ and $\text{Fit}(A, B) = \text{OLS}(A, B) := A^\dagger B$ is ordinary-least squares (OLS). The design matrices $(X_n)_{n \geq 0}$ are of shapes $T_n \times d$, each with iid rows from $N(0, \Sigma)$.

The sequence of noise vectors $(E_n)_{n \geq 0}$ forms an independent collection, which is independent of the $(X_n)_{n \geq 0}$; each $E_n \in \mathbb{R}^{T_n}$ has iid components $\epsilon_{n,i}$ from $N(0, \sigma_n^2)$. Refer to Figure 3.

---

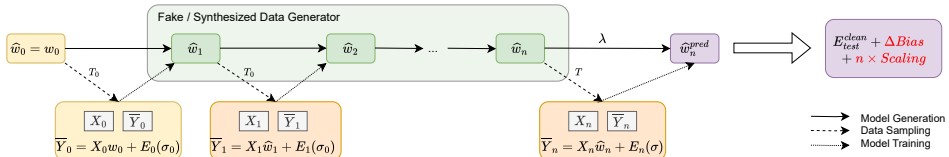

Figure 3: **Illustration of the theoretical framework.** The process begins with the original model $\widehat{w}_0(w_0)$ and the original dataset $(X_0, \overline{Y}_0)$. $n$ synthetic data generators $\widehat{w}_1$ to $\widehat{w}_n$ are iteratively fit on data labelled by the previous model with label noise $\sigma_0$, using $T_0$ samples each. We evaluate the test error (with respect to the ground truth labels from $w_0$) of $\widehat{w}_n^{pred}$, trained on $(X, Y) := (X_n, \overline{Y}_n)$ using $T$ samples with label noise $\sigma$ and a regularization coefficient $\lambda$.

Thus, in summary, each $\widehat{w}_n$ results from fitting a model on a dataset of size $T_{n-1}$ from $P_{\Sigma, \widehat{w}_{n-1}, \sigma_{n-1}^2}$, for every generation index $n \geq 1$.

**The Downstream Model: Ridge Regression.** For a number of iterations $n \geq 0$, noise levels $\sigma_0$ and $\sigma$, dataset sizes $T_0$ and $T$, and regularization parameter $\lambda \geq 0$, let $\widehat{w}_n^{pred} = \widehat{w}_{n, T_0, \sigma_0^2, T, \sigma, \lambda}^{pred} \in \mathbb{R}^d$ be the ridge predictor constructed from an iid sample $\{(x_1, y_1), \ldots, (x_T, y_T)\}$ of size $T$ from the $n$-fold fake data distribution $P_{\Sigma, \widehat{w}_n, \sigma_n^2}$, where for ease of presentation of our results we will assume that

$$T_{n-1} = \ldots = T_1 = T_0, \quad T_n = T \quad \text{and} \quad \sigma_{n-1} = \ldots = \sigma_1 = \sigma_0, \quad \sigma_n = \sigma. \tag{6}$$

For an $n$-fold fake data generator $P_{\Sigma, \widehat{w}_n, \sigma_n^2}$, we denote with $X := X_n \in \mathbb{R}^{T \times d}$ the design matrix with iid rows from $N(0, \Sigma)$, with $E := E_n \in \mathbb{R}^T$ the stage-$n$ label-noise vector with components in $N(0, \sigma_n^2)$, and $Y := \overline{Y}_n = X \widehat{w}_n + E \in \mathbb{R}^T$ the labels generated by $P_{\Sigma, \widehat{w}_n, \sigma_n^2}$. Let $\widehat{\Sigma} := X^\top X / T \in \mathbb{R}^{d \times d}$ is the sample covariance matrix, and $R = R(\lambda) := (\widehat{\Sigma} + \lambda I_d)^{-1}$ denote its *resolvent*, so that

$$\widehat{w}_n^{pred} = R X^\top Y / T \text{ for } \lambda > 0, \text{ and } \widehat{w}_n^{pred} = X^\dagger Y \text{ for } \lambda = 0. \tag{7}$$

We are interested in the dynamics of the test error $E_{test}(\widehat{w}_n^{pred})$ of this linear model. Importantly, the evaluation of the model is performed on the true data distribution $P_{\Sigma,w_0,\sigma_0^2}$, even though the model is trained on the fake data distribution $P_{\Sigma,\widehat{w}_n,\sigma^2}$. Note that for $n = 0$, $E_{test}^{clean} := E_{test}(\widehat{w}_n^{pred})$ corresponds to the usual test error when the downstream model is trained on clean data. Importantly, the downstream model has no control over this process. It will only see training data from a given version $P_{\Sigma,\widehat{w}_n,\sigma_n^2}$, but evaluation will be on the true distribution $P_{\Sigma,w_0,\sigma_0^2}$.

The mental picture is as follows: each generation $\widehat{w}_n$ can be seen as a proxy for a specific version of ChatGPT, for example. The sample size $T_0$ used to create the fake labelling functions $\widehat{w}_n$ is a proxy for the strength of the fake data-generator thus constructed. Other works which have considered model collapse under such a self-looping training process include [44, 2, 7, 18].

## 4 Exact Test Error Characterization

In this section we establish generic analytic formulae for the test error of the downstream model $\widehat{w}_n^{pred}$ (7) trained on $n$-fold fake data-generation as outlined in Section 3. The fully general technical key Theorem F.1 detailing formula (1), with a trace expression for $\rho$, (as well as proofs) are given in Appendix F; consult part F.1 for an exposition. Notations are standard (summarized in Appendix E).

### 4.1 Warm-up: Ordinary Least Squares on Isotropic Data

For a start, let us first consider the case of unregularized regression, where $\lambda = 0$ in Equation (7).

**Theorem 4.1.** *For an $n$-fold fake data generation process with $T_0 \geq d + 2$ samples, the test error for the linear predictor $\widehat{w}_n^{pred}$ in Equation* (7) *learned on $T \geq d + 2$ samples, with $\lambda = 0$, is given by*

$$E_{test}(\widehat{w}_n^{pred}) \simeq \frac{\sigma^2\phi}{1-\phi} + \frac{n\sigma_0^2\phi_0}{1-\phi_0}, \text{ with } \phi = \frac{d}{T}, \ \phi_0 = \frac{d}{T_0}, \tag{8}$$

*where the notation $f(T) \simeq g(T)$ means $f(T)/g(T) \to 1$, for large $T$.*

The first term $E_{test}(\widehat{w}_0^{pred}) \simeq \sigma^2\phi/(1-\phi)$ in the above decomposition corresponds to the usual error when the downstream model is fitted on clean data (see [22], for example). The additional term $n\sigma_0^2\phi_0/(1-\phi_0)$, proportional to the number of generations $n$, is responsible for model collapse.

**Model collapse versus more training data.** Note that the linear degeneration in test error highlighted by Equation (8) is a direct consequence of using the same dataset size $T_0$ across the fake data generator. Of course, if the underlying synthetic generating process has access to a larger data budget across generations, this decay can be significantly alleviated. For instance, if fake data increases gradually with the number of generations $m \geq 2$ as $T_m = (m\log^2 m)T_0$ (and, to simplify, $\sigma = \sigma_0$) a trivial extension of Theorem 4.1 yields

$$E_{test}(\widehat{w}_n^{pred}) \simeq (1 + \frac{1}{2\log^2 2} + \frac{1}{3\log^2 3} + \dots)E_{test}(\widehat{w}_0^{pred}) \simeq E_{test}(\widehat{w}_0^{pred}),$$

which will keep collapse at bay at the expense of largely increased training data ([44] also has a similar formula). This does not avoid model collapse; rather, it trades additional data generation and training effort against deterioration from generations of fake data. Thus, while for clean data increasing the dataset size n-fold leads to better scaling, with synthetic data, we forfeit this improvement. Also, note that we do not assume access to samples from any of the intermediate generation steps $\widehat{w}_0, \dots, \widehat{w}_{n-1}$; we only train the downstream model $\widehat{w}_n^{pred}$ on data from the last step $\widehat{w}_n$.

**Model Collapse as Change of Scaling Laws.** In the low-dimensional regime (fixed $d$), Theorem 4.1 already predicts a change of scaling law from $\sigma^2 T^{-1}$ to $\sigma^2 T^{-1} + n\sigma_0^2 T_0^{-1}$. Thus, as the sample size $T$ is scaled up, the test error eventually plateaus at the value $n\sigma_0^2 T_0^{-1}$ and does not vanish. This phenomenon, also established in [18] in the context of large language models, is clearly visible in Figure 1a. In the rest of this section and also in Section 5, we shall establish an analogous picture for high-dimensional regimes ($d \to \infty$).

**Mitigation via Regularization.** Note that the test error of the null predictor $w_{null} = 0$ is $E_{test}(w_{null}) = \|w_0\|_\Sigma^2$, and so

$$\frac{E_{test}(\widehat{w}_n^{pred})}{E_{test}(w_{null})} = \frac{1}{\text{SNR}}\frac{\phi}{1-\phi} + \frac{n}{\text{SNR}_0}\frac{\phi_0}{1-\phi_0},$$

where SNR $:= \|w_0\|_\Sigma^2/\sigma^2$ and SNR$_0 := \|w_0\|_\Sigma^2/\sigma_0^2$. We deduce that if $n \gg \text{SNR}_0/(1/\phi_0 - 1)$, then the learned model is already much worse than the null predictor! This suggests that a possible strategy for mitigating the negative effects on learning on AI-generated data is regularization, as empirically illustrated in Figures 1a, 1b, 2, and also in 4 of Appendix D.

Furthermore, in Section 5 we shall establish that the optimal regularization parameter established in [14], in the case of polynomially decreasing spectra (a regime which is relevant to wide neural networks), must be modified in the presence of synthetic training data in order to prevent the generalization error to diverge to infinity (i.e catastrophic failure).

## 4.2 High-Dimensional Regimes

In order to analyze the trace term $\rho$ appearing in Equation (1) (and spelled out in (32) in Appendix F.1), we need some tools from RMT, and ultimately obtain analytic formulae for $E_{test}(\widehat{w}_n^{pred})$ in Theorem 4.3. Such tools have been used extensively to analyze anisotropic ridge regression [41, 22, 4].

**Random Matrix Equivalents.** For any sample size $T \geq 1$ and $\lambda \geq 0$, define $\kappa(\lambda, T)$ implicitly by

$$\kappa(\lambda, T) - \lambda = \kappa(\lambda, T) \cdot \text{df}_1(\kappa(\lambda, T))/T, \tag{9}$$

where, for any $\lambda \geq 0$ and $m \in \mathbb{N}_\star$, $\text{df}_m(\lambda)$ is the $m$th order "degree of freedom" of the covariance matrix $\Sigma$ is given by $\text{df}_m(\lambda) = \text{df}_m(\lambda; \Sigma) := \text{tr}\,\Sigma^m(\Sigma + \lambda I_d)^{-m}$.

The effect of ridge regularization at level $\lambda \geq 0$ is to improve the condition of the empirical covariance matrix $\widehat{\Sigma}$; what the $\kappa$-function does is translate this into regularization on $\Sigma$ at level $\kappa(\lambda, T)$, so as control the capacity of the former, i.e. the "effective dimension" of the underlying problem. Quantitatively, there is an equivalence of the form $\text{df}_1(\lambda; \widehat{\Sigma}) \approx \text{df}_1(\kappa(\lambda, T); \Sigma)$. Roughly speaking, RMT is the business of formalizing such a relationship and derivatives (w.r.t. $\lambda$) thereof. A standard reference on the subject is [5].

**Example: Isotropic Data.** As an illustration, note that $\text{df}_m(\lambda) \equiv d/(1 + \lambda)^m$ (polynomial decay) in the isotropic case where $\Sigma = I_d$. Consequently, we have

$$\kappa(\lambda, T) - \lambda = \phi \cdot \kappa(\lambda, T)/(1 + \kappa(\lambda, T)), \text{ with } \phi := d/T.$$

In this case, it is easy to obtain the following well-known formula for $\kappa = \kappa(\lambda, T)$:

$$\kappa = \frac{\lambda + \overline{\phi} + \sqrt{(\lambda + \overline{\phi})^2 + 4\lambda}}{2}, \text{ with } \overline{\phi} := \phi - 1, \tag{10}$$

which is reminiscent of the celebrated Marchenko-Pastur law [34].

**Asymptotic Regime.** We shall work in the following so-called proportionate asymptotic scaling regime which is a standard analysis based on random matrix theory (RMT):

$$T, d \to \infty, \quad d/T \to \phi, \quad \|\Sigma\|_{op}, \|\Sigma^{-1}\|_{op} = O(1). \tag{11}$$

Later in Section 5 when we consider power-law spectra, this scaling will be extended to account for the more realistic case where $d$ and $T$ are of the same order on log scale, i.e

$$T, d \to \infty, \quad d^{1/C} \lesssim T \lesssim d^C, \quad \|\Sigma\|_{op}, \|\Sigma^{-1}\|_{op} = O(1), \tag{12}$$

for some absolute constant $C \geq 1$. Such non-proportionate settings are covered by the theory developed in [27, 48]. For clarity of presentation, even in this more general regime of Equations (12), we will still continue to write $\phi_0 := d/T_0$ and $\phi := d/T$.

**Bias-Variance Decomposition.** With everything now in place, let us recall for later use, the classical bias-variance decomposition for ridge regression (for example, see [41, 22, 4]):

**Proposition 4.2.** *In the RMT limit* (12)*, the test error of a ridge predictor $w(\lambda)$ based on $T$ iid samples from the true data distribution $P_{\Sigma, w_0, \sigma^2}$ is given by*

$$E_{test}(w(\lambda)) = \mathbb{E}\,\|w(\lambda) - w_0\|_\Sigma^2 \simeq Bias + Var, \tag{13}$$

$$\text{with } Bias \simeq \frac{\kappa^2 w_0^\top \Sigma(\Sigma + \kappa I)^{-2} w_0}{1 - \text{df}_2(\kappa)/T}, \quad Var \simeq \frac{\sigma^2 \text{df}_2(\kappa)}{T} \cdot \frac{1}{1 - \text{df}_2(\kappa)/T}, \tag{14}$$

*where $\kappa = \kappa(\lambda, T)$ is as given in Equation* (9)*.*

## 4.3 Analytic Formula for Test Error

The following result gives the test error for the downstream ridge predictor $\widehat{w}_n^{pred}$ defined in Equation (7), in the context of fake training data, and will be heavily exploited later to obtain precise estimates in different regimes. Define generic $Var$ and $Bias$ by:

$$Var = \mathbb{E}\,\|RX^\top E/T\|_{\widehat{\Sigma}}^2 = \sigma^2 \frac{1}{T}\operatorname{tr}\Sigma R^2\widehat{\Sigma} \qquad Bias = \mathbb{E}\,\|\widehat{\Sigma}Rw_0 - w_0\|_{\widehat{\Sigma}}^2,$$

and note that $E_{test}^{clean} := Bias + Var$, for standard ridge regression fitted on clean data from the true data distribution $P_{\Sigma,w_0,\sigma^2}$ (e.g., see Hastie et al. [22]). Let $Q_{n-1} = P_{n-1}P_{n-2}\ldots P_0$ where $P_m$ is the orthogonal projection unto the subspace of $\mathbb{R}^d$ spanned by the rows of $X_m$ and define

$$\Delta Bias \quad := \mathbb{E}\,\|\widehat{\Sigma}R(Q_{n-1}w_0 - w_0)\|_{\widehat{\Sigma}}^2 \geq 0. \tag{15}$$

**Theorem 4.3.** *For an $n$-fold fake data-generation process, the test error of a ridge predictor $\widehat{w}_n^{pred}$ based on a sample of size $T$ with regularization parameter $\lambda$, is given in the RMT limit (12) by*

$$E_{test}(\widehat{w}_n^{pred}) \simeq \widetilde{Bias} + Var + n\sigma_0^2\rho, \tag{16}$$

*where $\rho$ is as given in Theorem F.1, and $\widetilde{Bias}$ satisfies*

$$\widetilde{Bias} \geq Bias + \Delta Bias \geq Bias \text{ (with equality if } T_0 \geq d\text{)},$$

*and $\Delta Bias$ as given in (15). Furthermore, if one of the following conditions holds*

$$T_0 \geq d \ \ OR \ \ X_n = X_0 \text{ for all } n \geq 1, \tag{17}$$

*then, we have the following explicit formula for $\rho$*

$$\rho = \frac{\operatorname{tr}\Sigma^4(\Sigma + \kappa_0 I)^{-2}(\Sigma + \kappa I)^{-2}}{T_0 - \mathrm{df}_2(\kappa_0)} + \frac{\kappa^2\operatorname{tr}\Sigma^2(\Sigma + \kappa_0 I)^{-2}(\Sigma + \kappa I)^{-2}}{T_0 - \mathrm{df}_2(\kappa_0)} \cdot \frac{\mathrm{df}_2(\kappa)}{T - \mathrm{df}_2(\kappa)}, \tag{18}$$

*with $\kappa = \kappa(\lambda, T)$ and $\kappa_0 := \kappa(0, T_0)$ are as given in Equation* (9).

Instructively, the term $\Delta Bias$ measures how biased the synthetic data-generation process away from ground-truth model $w_0$. This term disappears if the generator was fitted on sufficiently many samples (i.e. if $T_0 \geq d$). More quantitatively, when $T_0 < d$ and $X_n = X_0$, it is easy to see that $\Delta Bias \geq \mathbb{E}\,[\|\Sigma^{1/2}\widehat{\Sigma}R\|_{op}^2] \cdot Bias_0$, where $Bias_0 := \mathbb{E}\,\|P_0 w_0 - w_0\|_2^2$ measures the inability due to lack of enough data, of the first generation ($n = 1$) to reliably estimate $w_0$ even in the absence of noise ($\sigma_0 = 0$) in the data-generating process. This gap propagates over to higher generations of the process. The situation is illustrated in Figure 2. In the case where $T_0 < d$ and the $X_n$'s are independent, we shall see in Section 4.5 that this increase in bias actually grows with $n$, even in the case of fake data generation without label noise (i.e. $\sigma_0 = 0$).

## 4.4 Model Collapse in the Case of Under-Parametrized Fake Data-Generator

We now consider the scenario of under-parameterization, where $T_0 \geq d$, indicating that the number of data points exceeds the number of dimensions. This condition typically results in a unique solution for the regression. In this case, $P_0 = I_d$ a.s., leading to $\widetilde{Bias} = Bias$ (given as in formula (14)), and $\kappa_0 = 0$ in (18), and so Theorem 4.3 gives

$$\rho = \frac{\mathrm{df}_2(\kappa)}{T_0 - d} + \frac{\kappa^2\operatorname{tr}(\Sigma + \kappa I)^{-2}}{T_0 - d}\frac{\mathrm{df}_2(\kappa)}{T - \mathrm{df}_2(\kappa)}. \tag{19}$$

We have the following corollary to Theorem 4.3.

**Corollary 4.4.** *Consider the setting of Theorems 4.3 and F.1. If $T_0 \geq d$ additionally, then it holds in the RMT limit* (12) *that $E_{test}(\widehat{w}_n^{pred}) \simeq Bias + Var + n\sigma_0^2\rho$, where $Bias$ and $Var$ are as given in formula* (14), *and $\rho$ is as given in Equation* (19).

*Moreover, in the special case of isotropic features, it holds that*

$$Bias + Var \simeq \frac{\kappa^2\|w_0\|_2^2 + \sigma^2\phi}{(1+\kappa)^2 - \phi}, \quad \rho \simeq \frac{\phi_0}{1 - \phi_0}\left(\frac{1}{(1+\kappa)^2} + \frac{1}{(1+\kappa)^2}\frac{\phi\kappa^2}{(1+\kappa)^2 - \phi}\right),$$

*with $\phi := d/T$, $\phi_0 := d/T_0$, and $\kappa = \kappa(\lambda, T)$ as in Equation* (10).

Note that Theorem 4.1 is special case of the above result corresponding to $\lambda = 0$ and $\phi \geq 1$. A result like Corollary 4.4 gives us the needed analytical handle for understanding $n$-fold model collapse in terms of all problem hyper-parameters (covariance spectrum, regularization, label-noise level, etc.).

### 4.5 Model Collapse in the Absence of Label Noise

We now consider the over-parametrized regime, where the different iterations of the synthetic data-generator (refer to the illustration in Figure 3) are fitted on insufficient data. For simplicity of exposition, we restrict our presentation to isotropic covariance $\Sigma = I_d$. Since we will be focusing on the possible increase $\Delta Bias$ above the bias (defined in Equation (14)) due to $n \geq 1$ generations as predicted by Theorem 4.3, we further restrict ourselves to the noiseless regime where the fake data-generating process has no label noise, i.e. $\sigma_0 = 0$. Thanks to Lemma F.4, we know that the generation-$n$ fake labelling vector $\widehat{w}_n$ (defined in Eqn. (5)) is given explicitly as a series of projections

$$\widehat{w}_n = Q_{n-1}w_0 = P_{n-1}P_{n-2}\ldots P_0 w_0. \tag{20}$$

Further, for simplicity we will assume $T = T_n > d$, i.e the downstream model has access to enough data. We shall focus on two important special cases.

**The Dependent Case.** We first consider the case where $T_m = T_0 < d$ and $X_m = X_0$ for all $m \leq n - 1$. It is clear that Equation (20) reduces to $\widehat{w}_n = P_0 w_0$, with $\operatorname{rank} P_0 = T_0 < d$.

**Theorem 4.5.** *In the limit $\lambda \to 0^+$ and $d, T_0 \to \infty$ with $d/T_0 \to \phi_0 > 1$, it holds that*

$$\|\widehat{w}_n\|^2 \simeq \|w_0\|^2/\phi_0, \quad \Delta Bias \simeq \|w_0\|^2(1 - 1/\phi_0). \tag{21}$$

We see that in this setting, the increase in bias $\Delta Bias \simeq (1-1/\phi_0)\|w_0\|^2$ brought about by synthetic data is a positive constant which does not grow with the number of generations $n \geq 1$. This increase in bias (i.e compared to training on clean data) is due to the fact that, with probability 1, the random subspace of $\mathbb{R}^d$ spanned by $X_0$ does not contain the ground truth model $w_0$. The expression is nothing but a RMT estimate of $\|P_0 w_0 - w_0\|^2$, i.e. the squared norm of the projection of $w_0$ onto the orthogonal complement of this subspace. The result is illustrated in Figure 2(a).

**The Independent Case.** For our second example, we remove the assumption that $T_m = T_0$ and $X_m = X_0$ for all $m \leq n - 1$ considered in the previous case (Theorem 4.5). We instead assume that (A) The $X_m$'s are assumed to be independent, and (B) we are in the following high-dimensional limit

$$\lambda \to 0^+, \quad d, T_1, \ldots, T_{n-1} \to \infty, \quad d/T_m \to \phi_m, \quad \text{for some } \phi_1, \ldots, \phi_{n-1} > 0. \tag{22}$$

Define $\eta := \prod_{m=0}^{n-1} \min(1/\phi_m, 1) \in (0, 1]$. We have the following theorem.

**Theorem 4.6.** *In the limit (22), it holds that $\|\widehat{w}_n\|^2 \simeq \|w_0\|^2 \eta$ and $\Delta Bias \simeq \|w_0\|^2 (1 - \eta)$. In particular, if $n \to \infty$ with infinitely many $\phi_m > 1$, then $\widehat{w}_n \to 0$ and $\Delta Bias \to \|w_0\|^2$.*

The theorem predicts that a sequence of over-parametrized fake data-generators $(\widehat{w}_n)_n$ collapses to zero (and thus, effectively escapes from the ground truth model $w_0$). Consequently, the downstream model $\widehat{w}_n^{pred}$ convergences to a Gaussian process around zero, instead of the true model $w_0$, leading to an increase in the bias term of the test error!

For example if $\phi_n = \phi_0 > 1$, then Theorem 4.6 predicts that $\Delta Bias \simeq (1 - \phi_0^{-n})\|w_0\|^2$, which grows exponentially fast towards $\|w_0\|^2$, the test error of the null predictor. This compounding effect is due to the fact that in (20), each projection $P_m$ spins the fake data labelling vector $\widehat{w}_n$ further away from the ground-truth $w_0$. The result is illustrated in Figure 2(b).

Comparing the dependent case and the independent case, 4.6 shows that the increase in bias is proportional to $1 - \eta_0 \eta_1 \ldots \eta_{n-1}$, which is typically much larger than $1 - \eta_0$, which is the increase in the dependent case. Sampling different design matrices results in a more pronounced model collapse.

## 5 The Case of Heavy Tails (Power Law)

Neural scaling laws [26, 24], relate a model's test error to the sample size, model size, and computational resources, and are critical tools for practitioners in strategically allocating resources during the design and implementation of large language models. Previous theoretical works [11, 41, 14] have examined scaling laws in our tractable setting of linear regression with Gaussian design in the context of a power-law covariance spectrum. Now we explore how synthetic data alters these scaling laws in this setting.

Let the spectral decomposition of the covariance matrix $\Sigma$ be $\Sigma = \lambda_1 v_1 v_1^\top + \ldots + \lambda_d v_d v_d^\top$, where $\lambda_1 \geq \ldots \geq \lambda_d \geq 0$ are the eigenvalues and $v_1, \ldots, v_d \in \mathbb{R}^d$ are the eigenvectors. For any feature index $j \in [d]$, define a coefficient $c_j := w_0^\top v_j$, i.e the projection of $w_0$ along the $j$th eigenvector of $\Sigma$. We shall work under the following well studied spectral conditions

$$\left.\begin{array}{l} \textbf{(Capacity Condition) } \lambda_j \asymp j^{-\beta} \text{ for all } j \in [d], \\ \textbf{(Source Condition) } \|\Sigma^{1/2-r} w_0\| = O(1), \end{array}\right\} \tag{23}$$

where $\beta > 1$ and $r > 0$. The parameter $r$ measures the amount of dispersion of $w_0$ relative to the spectrum of $\Sigma$; a large value of $r$ means $w_0$ is concentrated only along a few important eigen-directions (i.e. the learning problem is easy). For later convenience, define $\delta$, $\underline{r}$, and $c$ by

$$\delta := 1 + \beta(2r-1) \in \mathbb{R}, \quad \underline{r} := \min(r, 1) \in (0, 1), \quad c := 2\beta\underline{r}/(2\beta\underline{r}+1) \in (0, 1). \tag{24}$$

As noted in [14], the source condition in (23) is satisfied if $c_j \asymp j^{-\delta/2}$ for all $j \in [d]$.

Consider adaptive ridge regularization strength of the form

$$\lambda = \lambda(T) \asymp T^{-\ell}, \tag{25}$$

for fixed $\ell \geq 0$. The case where $\ell = 0$ corresponds to non-adaptive regularization; otherwise, the level of regularization decays polynomially with the sample size $T$. Define

$$\ell_{crit} := \beta/(2\beta\underline{r}+1). \tag{26}$$

In [14], KRR under normal circumstances (corresponding to $n = 0$, i.e. no fake data) was considered and it was shown that this value for the regularization exponent in (25) is minimax-optimal for normal test error in the noisy regime ($\sigma > 0$), namely $E_{test}(\widehat{w}_0^{pred}) \asymp T^{-c}$. This represents a crossover from the noiseless regime where it was shown that the test error scales like $E_{test}(\widehat{w}_0^{pred}) \asymp T^{-2\beta\underline{r}}$, a much faster rate. In the context of training on fake data, which is the object of this manuscript, we shall establish new scaling laws which paint a drastically different picture.

**A "Collapsed" Scaling Law.** The following result shows that model collapse is a modification of usual scaling laws induced by fake data. All proofs of this section can be found in Appendix H. Here, for simplicity of presentation, we restrict to the case $T_0 \geq d + 2$ to make the results easier to present. This condition can be removed as in Theorem 4.3.

**Theorem 5.1.** *Consider $n$-fold fake-data generation with sample size $T_0 \geq d + 2$. For a ridge predictor $\widehat{w}_n^{pred}$ given in Equation (7) based on a fake data sample of size $T$, with regularization parameter $\lambda = \lambda(T)$ tuned adaptively as in Equation (25) with exponent $\ell \in [0, \beta)$, the test error satisfies the following scaling law in the RMT limit (12):*

$$E_{test}(\widehat{w}_n^{pred}) \asymp \max(\sigma^2, T^{1-2\underline{r}\ell - \frac{\ell}{\beta}}) \cdot T^{-(1-\frac{\ell}{\beta})} + \frac{n\sigma_0^2}{1-\phi_0} \max(T/T_0, \phi_0) \cdot T^{-(1-\frac{\ell}{\beta})}. \tag{27}$$

We now provide an instructive interpretation of Theorem 5.1 and outline the effect of regularization.

**The Noiseless Regime.** First consider the case $\sigma = 0$ (or equivalently, exponentially small in $T$) and $\phi_0 \in (0, 1)$ is fixed, and consider a number of generations $n$ such that $n\sigma_0^2 \asymp T^a$, where $0 \leq a \leq 1 - \ell/\beta \leq 1$. Note that $a = 0$ corresponds to a constant number of generations. Also take $T_0 = T^b$, for some constant $b \in (0, \infty)$. According to Theorem 5.1, if we want to balance out the model-collapsing negative effect of training on fake data, we should chose $\ell$ so as to balance the second term $n(T/T_0)T^{-(1-\ell/\beta)} = T^{-(b-\ell/\beta-a)}$ and the first term $T^{-2\ell\underline{r}}$. We have the following:

**Corollary 5.2.** *In the setting of Theorem 5.1 with $T_0 \asymp T^b$ and $n \asymp T^b$, the optimal exponent of the ridge regularization parameter in Equation (25) is $\ell = \ell_\star$, where*

$$\ell_\star = \min((b-a)\ell_{crit}, \beta), \tag{28}$$

*and $\ell_{crit}$ is as in Eqn. (26), with corresponding optimal test error $\inf_{\ell \geq 0} E_{test}(\widehat{w}_n^{pred}) \asymp T^{-(b-a)c}$.*

Observe that when $(b-a)c < 2\beta\underline{r}$, which is the case when $n = O(1)$, $r \geq 1$ and $b \leq a + 1$, this corresponds to the condition $T \gtrsim T_0$. The above result represents a crossover from the fast rate $E_{test}(\widehat{w}_0^{pred}) \asymp T^{-2\beta\underline{r}}$ in the case of training on clean data [14], to a much slower rate

$E_{test}(\widehat{w}_n^{pred}) \asymp T^{-(b-a)c}$, attained by the adaptive regularization $\lambda \asymp T^{-\ell_\star}$, which is optimal in this setting. Furthermore, in this setting if we still use $\lambda \asymp T^{-\ell_{crit}}$ as proposed in [14] in the clean data setting, Corollary 5.2 predicts that

$$E_{test}(\widehat{w}_n^{pred}) \gtrsim T^{-(b-\ell_{crit}/\beta-a)} = T^{-(c+b-a-1)},$$

which diverges to infinity if $b \geq a + 1 - c$. This is a catastrophic form of model collapse, and is empirically illustrated in Figures 1b and 4.

**The Noisy Regime.** This discussion can be found in Appendix G.

**Remark.** In all the analyses above, we quantitatively demonstrate how model collapse manifests as a *change in scaling laws* within a setting commonly used to understand scaling behavior in current foundation models [46, 13, 14]. Our results indicate that, in the presence of synthetic data, scaling laws with respect to dataset size slow down (i.e., exhibit smaller exponents), meaning a much larger sample size is needed to achieve the same reduction in test error as with real data. Furthermore, the optimal scaling law with synthetic data requires different regularization; the optimal settings for real data could lead to catastrophic model collapse. Related findings are reported in Dohmatob et al. [18] in the setting of discrete data for infinite memory models and their variants.

## 6 Experiments

To further support our theoretical findings, we conduct experiments using kernel ridge regression on the MNIST dataset [16], as detailed in Appendix D.2. Our experiments validate the theoretical predictions for both RBF and Polynomial kernels, demonstrating the parallels between linear regression and kernel regression, and highlighting the relevance of our theory to more complex settings.

We also explore the behavior of real neural networks by training two-layer networks in two different settings: fixing the first layer or training both layers (see Appendix D.3). Consistent with our theoretical insights, we observe a linear pattern of model collapse when the first layer is fixed. However, a more severe, nearly quadratic model collapse is observed when both layers are trained, with our theory providing a lower bound for this behavior. These results reinforce the ability of our theory to capture the dynamics of model collapse across varying complexities. Full experimental details and results are provided in Appendix D.

## 7 Concluding Remarks

As we navigate the "synthetic data age", our findings signal a departure from traditional test error rates (e.g. neural scaling laws), introducing novel challenges and phenomena with the integration of synthetic data from preceding AI models into training sets. Our work provides a solid analytical handle for demystifying the model collapse phenomenon as a modification of usual scaling laws caused by fake / synthesized training data.

On the practical side, our analysis reveals that AI-generated data alters the optimal regularization for downstream models and changes the scaling laws. Drawing from the insight that regularization mirrors early stopping [3], our study suggests that models trained on mixed real and AI-generated data may initially improve but later decline in performance (model collapse), necessitating early detection of this inflection point. To preserve model quality when scaling laws are altered, it is essential to employ data filtering and watermarking techniques to distinguish real data from synthetic content. Recent studies have also explored methods for data selection [19] and correction [20]. These observations prompt a re-evaluation of current training approaches and underscores the complexity of model optimization in the era of synthetic data.

## Acknowledgments

YF and JK are supported by the National Science Foundation under NSF Award 1922658. Part of this work was done while JK and YF were hosted by the Centre Sciences de Données at the École Normale Supérieure (ENS) in 2023/24, whose hospitality they gratefully acknowledge. This work was partially supported through the NYU IT High Performance Computing (HPC) resources, services, and staff expertise.

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

# Appendix / Supplementary Material for
# Model Collapse Demystified: The Case of Regression

## Contents

## A  Differences to Self-Distillation

An important point we wish to make is that the fake data generation process that we analyse should
not be confused with self-distillation as formulated in Mobahi et al. [36] for example. Our setting

is inspired by the model collapse phenomenon, where increasingly vast amounts of synthetic data generated by users is posted online, and will necessarily enter the training set of the next foundation model. In this case, we do not have ground truth labels, nor is the generation of synthetic data not controlled by us, but by other users. Therefore, we adopt the setting of solely synthetic labels with added noise.

Specifically, in our setup, at generation $n > 0$, we do not have access to the true labels $Y_0 = f_0(X) + noise$ for the training samples $X$, but rather to some $\hat{Y}_n = \hat{f}_n(X) + noise$, where $\hat{f}_n$ is an unknown function, which synthesizes fake labels iteratively; the integer $n$ is the number of iterations. In our work, we make the structural assumption that $\hat{f}_n$ is obtained by iterative / successive regressions on a true dataset $D_0 = (X_0, Y_0)$. We do not have any control over the creation of these labels, which is reflected by the noise injected at each stage.

In the *self-distillation* setting, the data generation process actually helps performance of the downstream model. The model has access to training labels from the true data distribution $Y$, but decides to fit a model on this data, and then use its outputs as the new labels $Y_n := F_n(X, Y)$, iterating this process possibly over severable steps. Thus, self-distillation has control over the data generating process, which is carefully optimized for the next stage training. Specifically, [36] study self-distillation in the same Gaussian regression model underlying our analysis, but in each distillation generation are able to tune the regularization parameter for downstream performance as a function of the original data labels (with the data being the same at each generation). In the setting of model collapse, there is no control over the data generation process, since it constitutes synthesized data which typically comes from the wide web.

Self-distillation for linear regression would amount to a very special instance of our analysis where (1) $X_0 = X_1 = \ldots = X_{n-1} = X_n = X$ and (2) $\sigma_0 = \ldots = \sigma_{n-1} = 0$. That is, there is exactly one design matrix which is used in the data generation process and in the downstream estimator, and also no additional source of label noise is present at the end of each generation.

In the general setup considered in our work, (1) is not imposed. We typically assume that $X_0, X_1, \ldots, X_{n-1}, X_n$ with $X_n = X$, are all independent random matrices. An exception is line 247 ("The Dependent Case") of Section 3.5, where we assume $X_m = X_0$ for all $m \leq n - 1$, and independent of $X_n = X$. That setup (considered for the purposes of showing that model collapse can still occur in the absence of label noise) also assumes $\sigma_m = 0$ for all $m$; the analytic picture which emerges (Theorem 3.5) is already drastically different from what one would get from self-distllation (corresponding to additional assumption that $X = X_0$).

## B    Related work on Kernel Ridge Regression with Gaussian Design

This model has been studied by a vast body of works. For example, Richards et al. [41], Hastie et al. [22], Bach [4] analyze the classical bias-variance decomposition of the test error for ridge regression in the high dimensional setting where dataset size and dimension diverge proportionately, using tools from Random Matrix Theory (RMT). In Section 4 we significantly extend this type of analysis to training on iteratively generated synthetic data. This model is also particularly attractive because it allows to analyze an important trade-off: the relative decay of the eigenvalues of the kernel (*capacity*) and the coefficients of the target function in feature space (*source*). Sizeable effort has been dedicated to characterize the influence on the decay rate of the test error as a function of these two relative decays (aka *power laws*) [11, 38, 6, 41, 46, 14, 15]. In Section 5 we extend these efforts, in particular based on works of Cui et al. [13, 14] which has given a full characterization of all regimes and test error decay that can be observed at the interplay of noise and regularization, characterizing a crossover transition of rates in the noisy setting. Our work uncovers fascinating new effects as a result of iterative training on synthetic data.

## C    Extension to Kernel Methods

Though we present our results in the case of linear regression in $\mathbb{R}^d$ for clarity, they can be rewritten in equivalent form in the kernel setting. Indeed, as in [11, 45, 14, 29], it suffices to replace $x$ with a feature map induced by a kernel $K$, namely $\psi(x) := K_x \in \mathcal{H}_K$. Here, $\mathcal{H}_K$ is the reproducing kernel Hilbert space (RKHS) induced by $K$. In the data distribution (2), we must now replace the Gaussian marginal distribution condition $x \sim N(0, \Sigma)$ with $\psi(x) \sim N(0, \Sigma)$. The ground-truth

labeling linear function in (2) is now just a general function $f_0 \in L^2$. The predictor (7) is then given by (*Representer Theorem*) $\widehat{f}_n^{pred}(x) := K(X, x)^\top \widehat{c}_n$, with $\widehat{c}_n = (G + \lambda T I_d)^{-1} Y \in \mathbb{R}^n$, where $K(X, x) := (K(x_1, x), \ldots, K(x_T, x))$, and $G = K(X, X) \in \mathbb{R}^{n \times n}$ is the Gram matrix.

# D    Details of Experiments

We perform the following experiments on both simulated and real data to empirically validate our theoretical results.

## D.1    Simulated Data

We consider ordinary / linear ridge regression in $\mathbb{R}^d$, for $d = 300$ and different structures for the covariance matrix $\Sigma$ of the inputs: isotropic (i.e $\Sigma = I_d$) and power-law (23), with $(\beta, r) = (2, 0.375)$. For each value of $n$ (the generation index), the fake data-generator is constructed according to the process described in (4). Then, for different values of $T$ (between 1 and $1000, 000$), a sample of size $T$ is drawn from this fake data-generator and then a downstream ridge model (7) is fitted. The test set consists of $100, 000$ clean pairs $(x, y)$ form the true data distribution $P_{\Sigma, w_0, \sigma^2}$. This experiment is repeated 10 times to generate error bars. The results for the isotropic setting are shown in Figure 1a and the results for the power-law setting are shown in Figure 1b. Figure 2 shows the over-parametrized setting.

## D.2    Real Data: Kernel Ridge Regression on MNIST

As in Cui et al. [14], Wei et al. [48] we consider a distribution on MNIST [16], a popular dataset in the ML community. The classification dataset contains $60, 000$ training and $10, 000$ test data points (handwritten), with labels from 0 to 9 inclusive. Like in Cui et al. [14], we convert the labels into real numbers (i.e a regression problem) as follows: $y = $ label mod $2 + $ noise , where the variance of the noise is $\sigma^2 = 1$ (for simplicity, we also set $\sigma_0^2 = 1$). The test set consists of $10, 000$ pairs $(x, y)$, with the labels $y$ constructed as described in the previous sentence. The fake data used for training is generated as in the previous experiment, but via kernel ridge regression (instead of least squares) with the RBF kernel (bandwidth $= 10^{-4}$) and the polynomial kernel (degree $= 5$, bandwidth $= 10^{-3}$). Note that it was empirically shown in Cui et al. [14] that these datasets verify (23) with $(\beta, r) \approx (1.65, 0.097)$ in the case of the aforementioned RBF kernel, and $(\beta, r) \approx (1.2, 0.15)$ in the case of the polynomial kernel. Then, for different values of $T$ (between 1 and 1000), a sample of size $T$ is drawn from this fake data-generator and then a downstream kernel ridge model is fitted. Each of these experiments are repeated 10 times to generate error bars (due to different realizations of label noise). The results are shown in Figure 4.

## D.3    Neural Networks on MNIST

We now further examine model collapse in two-layer neural networks on the MNIST dataset, beyond the linear setting and Gaussian data. We consider two scenarios:

- learning with a random features (RF) model, where the first layer is fixed randomly, and only the second layer is trained, and
- learning with a fully trainable neural network.

For the two-layer network with the first layer fixed, our theory predicts a linear increase in test error as a function of the number of iterations $n$. This is because such models belong to the linearized regimes as finite-width random feature models and can be approximated by kernel regression [25, 45]. For fully-trained neural networks, our theory does not directly apply. However, we anticipate that the general trends uncovered by our asymptotic theory will hold true — for example, more parameters are expected to lead to greater model collapse, as shown in Theorem 4.1.

Specifically, the models were trained using stochastic gradient descent (SGD) with a batch size of 128 and a learning rate of 0.1. We employed a regression setting where labels were converted to one-hot vectors, and the model was trained using mean squared error for 200 epochs to convergence. When generating the synthetic data, Gaussian label noise with a standard deviation of 0.1 is added. The test error is consistently evaluated on the test set using clean labels.

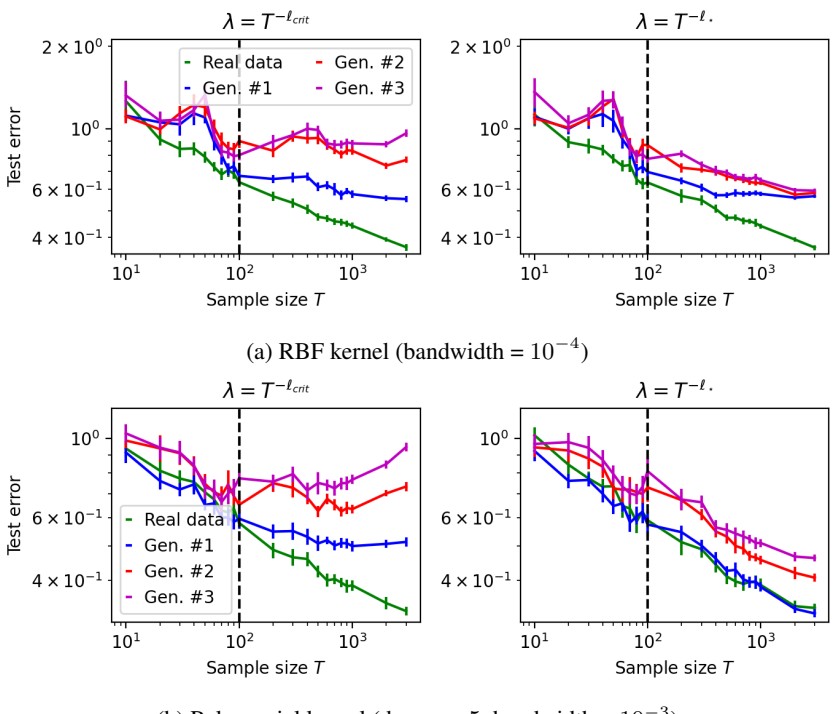

(a) RBF kernel (bandwidth = $10^{-4}$)

(b) Polynomial kernel (degree = 5, bandwidth = $10^{-3}$)

Figure 4: **Model collapse in kernel ridge regression (power-law covariance spectrum) on MNIST**. Here, we use adaptive regularization $T^{-\ell}$ for different values of the exponent $\ell \geq 0$ (see Section D for full experimental setup). **Top row:** RBF kernel. **Bottom row:** polynomial kernel. In each plot, we show test error curves as a function of sample size $T$, from different generations ($n$) of fake data. The broken vertical line corresponds to $T = T_0$, where $T_0$ is the number of samples (from the true data distribution) which was used to train the label faker. The value of the exponent regularization $\ell = \ell_\star$ (broken curves) is the optimal value in the presence of iterative data relabeling, while $\ell = \ell_{crit}$ (solid curves) corresponds to the optimal value without iterative re-labelling (i.e $n = 0$) proposed in Cui et al. [14] (see (26)). Specifically, we take $\ell_\star = (b-a)\ell_{cirt} = b\ell_{crit}$, where $b = \log T_0 / \log T$ (so that $T_0 = T^b$), as proposed in Theorem 5.1, formula (28). Notice how the effect of fake data makes the test error become non decreasing in sample size $T$. This is effectively a collapse of the learned model.

The results for RF models of width (i.e number of hidden dimensions) $k$ of 20,000 are presented in Figure 5. We observe that, with the exception of the first two generations, the decay in MSE loss generally follows a linear trend, which is consistent with the predictions of our theory.

Next, we consider the scenario of training the entire neural network. By varying the width $k$, we adjust the number of parameters to further explore the theoretical predictions on how the number of parameters influences model collapse.

**Observations.** From Figure 6, we can observe that

- More parameters (wider neural networks, i.e large $k$) lead to increased model collapse. This observation is consistent with our results proved in the linear regime (Theorem 4.1). For linear models, the number of parameters is proportional to $d$ (the input dimension), whereas in two-layer neural networks, the number of parameters is of order $kd$ (i.e proportional to the width $k$).

- The dependence of model collapse on the number of iterations $n$ is linear for small values of $n$ (with $n \leq 4$ in our experiments), and becomes superlinear (possibly quadratic) for larger values of $n$ (with $n \geq 4$). Recall that $n = 0$ corresponds to training on clean data from the data distribution. Thus, possibly, model collapse neural networks appears to be even more severe than in linear regression.

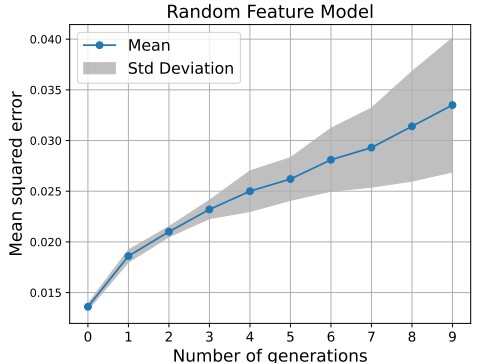
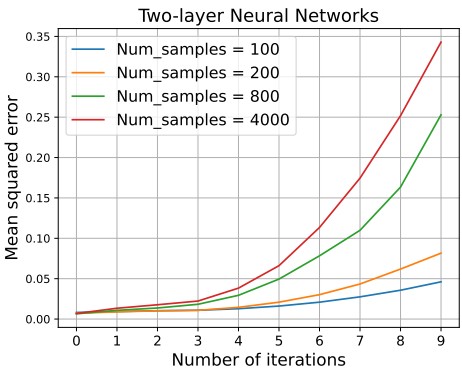

Figure 5: Performance of RF model on MNIST, with one-hidden layer NN (width $k = 20,000$). Standard deviation is calculated using 10 seeds.

Figure 6: The performance of two-layer neural network on MNIST with varying hidden dimensions.

## E  Notations

The set of integers from $1$ through $d$ is denoted $[d]$. Given a variable $z$ (which can be the input dimension $d$ or the sample size $T$, etc.) the notation $f(z) \lesssim g(z)$ means that $f(z) \leq Cg(z)$ for sufficiently large $z$ and an absolute constant $C$, while $f(z) \asymp g(z)$ means $f(z) \lesssim g(z) \lesssim f(z)$. Further, $f(z) \simeq g(z)$ means $f(z) = (1 + o(1))g(z)$, where $o(1)$ stands for a quantity which tends to zero in the limit $z \to \infty$. We denote with $A^\dagger$ the Moore-Penrose pseudo-inverse any matrix $A$, and by $\|A\|_{op}$ is operator norm, while the trace of a square matrix $A$ is denoted $\operatorname{tr} A$. Finally, $\|u\|_\Sigma := \sqrt{u^\top \Sigma u}$ is the Mahalanobis norm induced by a positive-definite matrix $\Sigma$.

## F  Exact Characterization of Test Error Under Model Collapse

### F.1  A General Formula for Test Error

We now consider the case of general ridge penalty $\lambda > 0$, and drop the requirements $T \geq d + 2$ and $T_0 \geq d + 2$. Recall the definitions of $X, Y, E$ and the random matrices $R$ and $\widehat{\Sigma}$ appearing in (7). For later reference, define

$$Bias := \mathbb{E}\,\|\widehat{\Sigma}Rw_0 - w_0\|_\Sigma^2, \tag{29}$$

$$Var = \mathbb{E}\,\|RX^\top E/T\|_\Sigma^2 = \sigma^2 \frac{1}{T}\operatorname{tr}\Sigma R^2\widehat{\Sigma}. \tag{30}$$

These are respectively the bias and variance terms in the classical bias-variance decomposition

$$E_{test}^{clean} := Bias + Var, \tag{31}$$

for standard ridge regression fitted on clean data from the true data distribution $P_{\Sigma, w_0, \sigma^2}$ (e.g., see Hastie et al. [22]).

**Theorem F.1.** *For an $n$-fold fake data generation process, the test error of a ridge predictor $\widehat{w}_n^{pred}$ based on a sample of size $T \geq 1$ with regularization parameter $\lambda$ is given by*

$$\left.\begin{aligned}
E_{test}(\widehat{w}_n^{pred}) &= \widetilde{Bias} + Var + n\sigma_0^2\rho, \\
\widetilde{Bias} &= \mathbb{E}\,\|\widehat{\Sigma}RQ_{n-1}w_0 - w_0\|_\Sigma^2, \\
\rho &:= \frac{1}{n}\sum_{m=0}^{n-1}\mathbb{E}\,\operatorname{tr} C_{n-1,m}\widehat{\Sigma}R\Sigma R\widehat{\Sigma},
\end{aligned}\right\} \tag{32}$$

*where $Var$ is as given in (30) and $C_{k,m} := \overline{Q}_{k,m}\overline{Q}_{k,m}^\top$ for $\overline{Q}_{k,m} = Q_{k,m}X_m^\dagger$, $Q_{k,m} := P_k P_{k-1}\dots P_m$, $Q_k := Q_{k,0} = P_k P_{k-1}\dots P_0$, with $P_m = X_m^\dagger X_m$ being the orthogonal projection matrix onto the subspace of $\mathbb{R}^d$ spanned by the rows of $X_m$.*

*In particular, if $T_0 \geq d + 2$ (under-parametrized data-generator), then $\widetilde{Bias} = Bias$ as in (29), and*

$$\left. \begin{aligned} E_{test}(\widehat{w}_n^{pred}) &\simeq E_{test}^{clean} + n\sigma_0^2\rho, \\ \rho &= \frac{1}{T_0 - d - 1}\mathbb{E}\operatorname{tr}\Sigma^{-1}\widehat{\Sigma}R\Sigma\widehat{\Sigma}R. \end{aligned} \right\}$$

(33)

In the second part of the theorem, the term $E_{test}^{clean}$ (introduced earlier in (31)) corresponds to the usual test error when the downstream model is trained on real (not fake) data, for which well-known formulae exist in a variety of scenarios (see Proposition 4.2).

**Remark F.2.** *We show in Theorem 4.3 that $\widetilde{Bias} \geq Bias + \Delta Bias$, where $\Delta Bias \geq 0$ in the appropriate asymptotic limit, with equality if $T_0 \geq d + 2$ (the under-parametrized regime). Thus, apart from the variance term, an over-parametrized ($T_0 < d + 2$) synthetic data-generator harms the bias term of the test error of downstream models. In contrast, an under-parametrized synthetic data-generator ($T_0 \geq d + 2$) only harms the variance. The increase in bias suffered in the over-parametrized regime is precisely quantified in Section 4.5, and shown to be an increasing function of the number of generations $n$.*

The test error decomposition in Theorem F.1 is thus of the promised form (1). This additional term means that there is competition between the usual test error $E_{test}^{clean}$ and the additional term induced by the fake labeling process. Understanding the interaction of these two terms is key to demystifying the origins of model collapse.

**Low-Dimensional Limit.** Observe that if $d$ is fixed and $T \to \infty$, then the empirical covariance matrix $\widehat{\Sigma}$ converges to[2] its population version $\Sigma$, and so for $T_0 \geq d + 2$, we have

$$\rho \simeq \frac{\operatorname{tr}\Sigma^2(\Sigma + \lambda I_d)^{-2}}{T_0 - d} = \frac{\mathrm{df}_2(\lambda)}{T_0 - d},$$

where for any $\lambda \geq 0$ and $m \in \mathbb{N}_\star$, $\mathrm{df}_m(\lambda)$ is the $m$th order "degree of freedom" of the covariance matrix $\Sigma$ is given by

$$\mathrm{df}_m(\lambda) = \mathrm{df}_m(\lambda; \Sigma) := \operatorname{tr}\Sigma^m(\Sigma + \lambda I_d)^{-m}.$$

Note that $\mathrm{df}_m(\lambda) \leq d$ always. In the high-dimensional setting (where $d$ can grow beyond $T_0$), the precise analysis of $\rho$ will be carried out via random matrix theory (RMT).

## F.2 Proof of Theorem 4.1 (Rigeless Regression)

The proof is by induction on the number of generations $n$ of fake data. For $n = 0$, we have

$$\begin{aligned} E_{test}(\widehat{w}_0^{pred}) &= \mathbb{E}\|\widehat{w}_0^{pred} - w_0\|_\Sigma^2 = \mathbb{E}\|\widehat{w}_0^{pred} - \widehat{w}_0\|_2^2 = \mathbb{E}\|(X_0^\top X_0)^{-1}X_0^\top E_0\|_2^2 \\ &= \sigma^2\mathbb{E}\operatorname{tr}(X_0^\top X_0)^{-1} = \sigma^2\frac{d}{T - d - 1} \simeq \frac{\sigma^2\phi}{1 - \phi}, \end{aligned}$$

(34)

where $\phi := d/T \in (0, 1)$ and the last step has made use of Lemma F.3 below. This is a well-known result for the test error of linear regression in the under-parametrized regime, without any AI pollution (fake / synthesized training data).

---

[2]e.g. weakly, w.r.t. operator-norm.

Analogously, for $n = 1$ one computes the test error after the first generation of fake data as follows

$$
\begin{aligned}
E_{test}(\widehat{w}_1^{pred}) &= \mathbb{E}\|\widehat{w}_1^{pred} - w_0\|_\Sigma^2 = \mathbb{E}\|\widehat{w}_1^{pred} - \widehat{w}_0\|_2^2 \\
&= \mathbb{E}\|\widehat{w}_1^{pred} - \widehat{w}_1 + \widehat{w}_1 - \widehat{w}_0\|_2^2 \\
&= \mathbb{E}\|(X_1^\top X_1)^{-1} X_1^\top E_1 + \widehat{w}_0^{pred} - \widehat{w}_0\|_2^2 \\
&= \mathbb{E}\|w_0 - \widehat{w}_0^{pred}\|_2^2 + \mathbb{E}\|(X_1^\top X_1)^{-1} X_1^\top E_1\|_2^2 \\
&= E_{test}(\widehat{w}_0^{pred}) + \frac{\sigma_1^2 d}{T_1 - d - 1} \\
&= E_{test}(\widehat{w}_0^{pred}) + \frac{\sigma_0^2 d}{T_0 - d - 1} \\
&\simeq \frac{\sigma^2 \phi}{1 - \phi} + \frac{\sigma_0^2 \phi_0}{1 - \phi_0},
\end{aligned}
$$

where $\phi_0 = d/T_0 \in (0, 1)$. Continuing the induction on $n$, we obtain the result. $\qquad\square$

**Lemma F.3.** *Let $X_0$ be an $T_0 \times d$ random matrix with iid rows from $N(0, \Sigma)$. If $T_0 \geq d + 2$, then the empirical covariance matrix $\widehat{\Sigma}_0 := X_0^\top X_0 / T_0$ is invertible a.s and*

$$
\mathbb{E}\left[\widehat{\Sigma}_0^{-1}\right] = \frac{T_0}{T_0 - d - 1} \Sigma^{-1}.
$$

## F.3 Proof of Theorem F.1 (Ridge Regression + General Covariance)

### F.3.1 Representation of $\widehat{w}_n$ and $\widehat{w}_n^{pred}$

We first obtain explicit formulae for the labelling vectors $\widehat{w}_n$ used in the fake-data generation process (5). For any integer $m \geq 0$, define $P_m = X_m^\dagger X_m$, the orthogonal projection matrix onto the subspace of $\mathbb{R}^d$ spanned by the rows of $X_m$. Observe from (5) that

$$
\begin{aligned}
\widehat{w}_n &= X_{n-1}^\dagger \overline{Y}_{n-1} = X_{n-1}^\dagger (X_{n-1} \widehat{w}_{n-1} + E_{n-1}) = P_{n-1} \widehat{w}_{n-1} + X_{n-1}^\dagger E_{n-1} \\
&= P_{n-1} X_{n-2}^\dagger (X_{n-2} \widehat{w}_{n-2} + E_{n-2}) + X_{n-1}^\dagger E_{n-1} \\
&= P_{n-1} P_{n-2} \widehat{w}_{n-2} + P_{n-1} X_{n-2}^\dagger E_{n-2} + X_{n-1}^\dagger E_{n-1} \\
&\ \ \vdots \\
&= P_{n-1} P_{n-2} \dots P_0 w_0 + P_{n-1} P_{n-2} \dots P_1 X_1^\dagger E_1 + P_{n-1} P_{n-2} \dots P_2 X_2^\dagger E_2 + \dots \\
&\ \ \vdots \\
&= P_{n-1} P_{n-2} \dots P_0 w_0 + \sum_{m=0}^{n-1} P_{n-1} P_{n-2} \dots P_m X_m^\dagger E_m.
\end{aligned}
\tag{35}
$$

We get the following result.

**Lemma F.4.** *For any $n \geq 0$, the following formula holds*

$$
\widehat{w}_n = \begin{cases} w_0, & \text{if } n = 0, \\ Q_{n-1} w_0 + \sum_{m=0}^{n-1} \overline{Q}_{n-1,m} E_m, & \text{if } n \geq 1, \end{cases}
\tag{36}
$$

*where $\overline{Q}_{k,m} := Q_{k,m} X_m^\dagger$, $Q_{k,m} := P_k P_{k-1} \dots P_m$ and $Q_k := Q_{k,0} = P_k P_{k-1} \dots P_0$. Moreover, $\widehat{w}_n \in \operatorname{Im} P_{n-1}$ as soon as $n \geq 1$.*

*In particular, under the simplifying condition* (17)*, it holds that*

$$
\widehat{w}_n = \begin{cases} w_0, & \text{if } n = 0, \\ P_0 w_0 + X_0^\dagger \overline{E}_{n-1} \in \operatorname{Im} P_0, & \text{if } n \geq 1. \end{cases}
\tag{37}
$$

*where $\overline{E}_{n-1} := \sum_{m=0}^{n-1} E_m$, a random vector of length $T_0$, with iid entries from $N(0, n\sigma_0^2)$, and independent of $X_0$. Moreover, $\widehat{w}_n \in \operatorname{Im} P_0$ as soon as $n \geq 1$.*

Note that the second part of the result uses the elementary linear-algebraic fact that $P_m X_m^\dagger = X_m^\dagger$. In the special case where $T_0 \geq d$, we have $P_0 = I$ a.s., and so $\widehat{w}_n = w_0 + X_0^\dagger \overline{E}_{n-1}$. Otherwise, even in the absence of generator noise ($\sigma_0 = 0$), the fake data labeller $\widehat{w}_n = P_0 w_0$ drifts away from the truth $w_0$, into a subspace of $\mathbb{R}^d$ spanned by the rows of $X_0$.

Next, let us obtain a decomposition for the downstream predictor $\widehat{w}_n^{pred}$ defined in (7). As usual, let $\widehat{\Sigma} := X^\top X / T$ be the empirical covariance matrix with resolvent $R = (\widehat{\Sigma} + \lambda I)^{-1}$, and observe that the downstream model writes

$$
\begin{aligned}
\widehat{w}_n^{pred} &= R X^\top \overline{Y}_n(X)/T = R X^\top (X \widehat{w}_n + E)/T \\
&= R X^\top (X Q_{n-1} w_0 + X \sum_{m=0}^{n-1} \overline{Q}_{n-1,m} E_m + E)/T \\
&= R \widehat{\Sigma} Q_{n-1} w_0 + R X^\top E / T + R \widehat{\Sigma} \sum_{m=0}^{n-1} \overline{Q}_{n-1,m} E_m.
\end{aligned}
\tag{38}
$$

### F.3.2   Proof of Theorem F.1

Using the decomposition (38) for the downstream model $\widehat{w}_n^{pred}$, we deduce that

$$
\begin{aligned}
E_{test}(\widehat{w}_n^{pred}) &= \mathbb{E} \, \|\widehat{w}_n^{pred} - w_0\|_\Sigma^2 \\
&= \mathbb{E} \, \|R \widehat{\Sigma} P_0 w_0 + R X^\top E/T + R \widehat{\Sigma} \sum_{m=0}^{n-1} \overline{Q}_{n-1,m} E_m - w_0\|_\Sigma^2 \\
&= \mathbb{E} \, \|R \widehat{\Sigma} P_0 w_0 - w_0 + R X^\top E/T + R \widehat{\Sigma} \sum_{m=0}^{n-1} \overline{Q}_{n-1,m} E_m\|_\Sigma^2 \\
&= \mathbb{E} \|R \widehat{\Sigma} P_0 w_0 - w_0\|_\Sigma^2 + \mathbb{E} \, \|R X^\top E/T\|_\Sigma^2 + \mathbb{E} \, \left\| R \widehat{\Sigma} \sum_{m=0}^{n-1} \overline{Q}_{n-1,m} E_m \right\|_\Sigma^2 \\
&= \widetilde{Bias} + Var + n\sigma_0^2 \rho,
\end{aligned}
\tag{39}
$$

where $\widehat{\Sigma} := X^\top X / T$, $\widetilde{Bias}$, $Var$, and $\rho$ are as given in the theorem. On the second line, we have used independence (of $X$, $X_0$, $E$, and $\overline{E}_{n-1}$) and the fact that $E$ and $\overline{E}_{n-1}$ are centered Gaussian random vectors, with iid components of variances $\sigma^2$ and $n\sigma_0^2$ respectively. $\qquad\square$

### F.4   Proof of Theorem 4.3

**Analysis of Bias-like Term.**   An exact analysis of the $\widetilde{Bias}$ term appearing in Theorems F.1 and 4.3 is presumably a treacherous enterprise given dependency on $X$ (via $R$ and $\widehat{\Sigma}$) and $X_0$ (via $P_0$). In place of such an analysis, we shall settle for the following result which gives an instructive lower-bound.

**Proposition F.5.** *In the RMT limit* (12)*, it holds that*

$$
\lim \widetilde{Bias} - \lim Bias \geq \lim \mathbb{E} \, \|R \widehat{\Sigma} P_0 w_0 - R \widehat{\Sigma} w_0\|_\Sigma^2 \geq 0.
$$

*Thus, training on fake / synthesized data increases the bias term of the downstream model's test error!*

*Proof.* Letting $A := R\widehat{\Sigma}$, one computes

$$
\begin{aligned}
\widetilde{Bias} - Bias &= \|A P_0 w_0 - w_0\|_\Sigma^2 - \|A w_0 - w_0\|_\Sigma^2 \\
&= \|A P_0 w_0 - A w_0 + A w - w\|_\Sigma^2 - \|A w_0 - w_0\|_\Sigma^2 \\
&= \|A P_0 w_0 - A w_0\|_\Sigma^2 + 2 w_0^\top (A - P_0 A) \Sigma (I - A) w_0 \\
&= \|A P_0 w_0 - A w_0\|_\Sigma^2 + 2 w_0^\top (I - P_0) A \Sigma (I - A) w_0.
\end{aligned}
$$

It then suffices to observe that, in the RMT limit (12), it holds that

$$
\lim \mathbb{E} \, w_0^\top (I - P_0) A \Sigma (I - A) w_0 \geq 0,
$$

as can be seen from repeated application of Propositions 1 and 2 of Bach [4]. $\qquad\square$

**Analysis of $\rho$ Term.** Define a $d \times d$ random psd matrix $H := \widehat{\Sigma} R \Sigma \widehat{\Sigma} R$. Under the simplifying assumption (17), the matrices $\overline{Q}_{k,m}$ defined in the theorem all equal $\overline{Q}_{0,0} = X_0^\dagger$. It follows that the $\rho$-term in (32) then writes

$$\rho = \frac{1}{n} \sum_{m=0}^{n-1} \mathbb{E}\left[\overline{Q}_{n-1,m} \overline{Q}_{n-1,m}^\top H\right] = \mathbb{E}\left[\operatorname{tr} X_0^\dagger (X_0^\dagger)^\top H\right] = \mathbb{E}_H \mathbb{E}\left[\operatorname{tr} X_0^\dagger (X_0^\dagger)^\top H \mid H\right]. \quad (40)$$

Now, one computes the conditional expectation as follows

$$\mathbb{E}\left[\operatorname{tr} X_0^\dagger (X_0^\dagger)^\top H \mid H\right] = \mathbb{E}\left[\operatorname{tr} X_0^\top (X_0 X_0^\top)^{-2} X_0 H \mid H\right]$$

$$= \lim_{\lambda_0 \to 0+} \frac{1}{T_0} \frac{\partial}{\partial \lambda_0} \mathbb{E}\left[\operatorname{tr} X_0^\top (X_0 X_0^\top + \lambda_0 T_0 I)^{-1} X_0 H \mid H\right].$$

Furthermore, defining $A := \Sigma^{1/2} H \Sigma^{1/2}$ and $Z_0 = X_0 \Sigma^{-1/2}$, we have

$$\operatorname{tr} X_0^\top (X_0 X_0^\top + \lambda_0 T_0 I)^{-1} X_0 H = \operatorname{tr} \Sigma^{1/2} Z_0^\top (Z_0 \Sigma Z_0^\top + \lambda_0 T_0 I)^{-1} Z_0 \Sigma^{1/2} H$$

$$= \operatorname{tr} A Z_0^\top (Z_0 \Sigma Z_0^\top + \lambda_0 T_0 I)^{-1} Z_0,$$

We deduce from Proposition 2 of Bach [4] that

$$\mathbb{E}\left[\operatorname{tr} X_0^\top (X_0 X_0^\top + \lambda_0 T_0 I)^{-1} X_0 H \mid H\right] \simeq \operatorname{tr} A(\Sigma + \kappa(\lambda_0, T_0) I)^{-1}$$

$$= \operatorname{tr} H(\Sigma + \kappa(\lambda_0, T_0) I)^{-1} \Sigma.$$

Differentiating w.r.t. $\lambda_0$ and letting this parameter tend to zero from above gives

$$\mathbb{E}\left[\operatorname{tr} X_0^\dagger (X_0^\dagger)^\top H \mid H\right] = -\frac{1}{T_0} \lim_{\lambda_0 \to 0+} \frac{\partial}{\partial \lambda_0} \mathbb{E}\left[\operatorname{tr} X_0^\top (X_0 X_0^\top + \lambda_0 T_0 I)^{-1} X_0 H \mid H\right]$$

$$\simeq -\frac{1}{T_0} \lim_{\lambda_0 \to 0+} \frac{\partial \kappa(\lambda_0, T_0)}{\partial \lambda_0} \cdot \frac{\partial}{\partial t} \operatorname{tr} H(\Sigma + tI)^{-1} \Sigma \Big|_{t=\kappa(\lambda_0, T_0)}$$

$$\simeq \frac{\operatorname{tr} H(\Sigma + \kappa_0 I)^{-2} \Sigma}{T_0 - \operatorname{df}_2(\kappa_0)},$$

where $\kappa_0 = \kappa(0, T_0)$, and we have made use of Lemma I.2. Combining with (40) and then applying Proposition 1 of Bach [4] to compute $\mathbb{E}_H \operatorname{tr} H(\Sigma + \kappa_0 I)^{-2} \Sigma = \mathbb{E}_X \operatorname{tr} \widehat{\Sigma} R \Sigma \widehat{\Sigma} R (\Sigma + \kappa_0 I)^{-2} \Sigma$ gives the following result.

**Proposition F.6.** *In the RMT limit* (12)*, it holds for any $\lambda > 0$ that*

$$\rho = \frac{\operatorname{tr} \Sigma^4 (\Sigma + \kappa_0 I)^{-2} (\Sigma + \kappa I)^{-2}}{T_0 - \operatorname{df}_2(\kappa_0)} + \frac{\kappa^2 \operatorname{tr} \Sigma^2 (\Sigma + \kappa_0 I)^{-2} (\Sigma + \kappa I)^{-2}}{T_0 - \operatorname{df}_2(\kappa_0)} \cdot \frac{\operatorname{df}_2(\kappa)}{T - \operatorname{df}_2(\kappa)}, \quad (41)$$

*where $\kappa_0 := \kappa(\lambda_0, T_0)$ and $\kappa = \kappa(\lambda, T)$.*

*In particular, if $T_0 \geq d$, then*

$$\rho \simeq \frac{\operatorname{df}_2(\kappa)}{T - \operatorname{df}_2(\kappa)} \left(1 + \frac{\kappa^2 \operatorname{tr}(\Sigma + \kappa I)^{-2}}{T_0 - \operatorname{df}_2(\kappa_0)}\right). \quad (42)$$

This result completes the proof of Theorem 4.3. $\qquad \square$

### F.5 Proof of Corollary 4.4

For the first part, we know from Theorem F.1 that

$$E_{test}(\widehat{w}_n^{pred}) = E_{test}(\widehat{w}_0^{pred}) + n\sigma_0^2 \rho, \quad \text{with} \quad (43)$$

$$\rho := \frac{\mathbb{E} \operatorname{tr} \Sigma^{-1} \widehat{\Sigma} (\widehat{\Sigma} + \lambda I)^{-1} \Sigma (\widehat{\Sigma} + \lambda I)^{-1} \widehat{\Sigma}}{T_0 - d}. \quad (44)$$

The $E_{test}(\widehat{w}_0^{pred})$ term is taken care of by Proposition 4.2, since this corresponds to generalization error on clean training data. For the $\rho$ term, we use Proposition 1 of Bach [4] with $A = \Sigma^{-1}$ and $B = \Sigma$ to get

$$
\begin{aligned}
\rho &\simeq \frac{\operatorname{tr}(\Sigma + \kappa I)^{-2}\Sigma^2}{T_0 - d} + \frac{\kappa^2 \operatorname{tr}(\Sigma + \kappa I)^{-2})}{T_0 - d} \frac{\operatorname{tr}(\Sigma + \kappa I)^{-2}\Sigma^2}{T - \mathrm{df}_2(\kappa)} \\
&= \frac{\mathrm{df}_2(\kappa)}{T_0 - d} + \frac{\kappa^2 \operatorname{tr}(\Sigma + \kappa I)^{-2}}{T_0 - d} \frac{\mathrm{df}_2(\kappa)}{T - \mathrm{df}_2(\kappa)},
\end{aligned}
$$

which proves the first part of the result.

For the second part, note that $\mathrm{df}_2(\kappa) = d/(1+\kappa)^2$ when $\Sigma = I$, (10) holds, and so

$$
\begin{aligned}
(1 - 1/\phi_0)\rho &\simeq \frac{1}{(1+\kappa)^2} + \frac{\kappa^2}{(1+\kappa)^4} \frac{d}{T - d/(1+\kappa)^2} \\
&\simeq \frac{1}{(1+\kappa)^2} + \frac{\kappa^2}{(1+\kappa)^4} \frac{\phi}{1 - \phi/(1+\kappa)^2} \\
&= \frac{1}{(1+\kappa)^2} + \frac{1}{(1+\kappa)^2} \frac{\phi\kappa^2}{(1+\kappa)^2 - \phi},
\end{aligned}
$$

and the result follows. $\qquad\square$

### F.6  A Note on Proposition 4.2

As mentioned in the main text, the result is classical Richards et al. [41], Hastie et al. [22], Bach [4]). Only the second part needs a comment which we now provide. Indeed, the second part of the result follows from the first as we now see. Indeed, $w_0^\top \Sigma(\Sigma + \kappa I)^{-2} w_0 = r^2/(1+\kappa)^2$, $\mathrm{df}_2(\kappa) = d/(1+\kappa)^2$ and so we deduce from the first part that

$$
\begin{aligned}
Var &\simeq \sigma^2 \phi \frac{1}{(1+\kappa)^2} \frac{1}{1 - \phi/(1+\kappa)^2} = \frac{\sigma^2 \phi}{(1+\kappa)^2 - \phi}, \\
Bias &\simeq \kappa^2 \|w_0\|_2^2 \frac{1}{(1+\kappa)^2} \frac{1}{1 - \phi/(1+\kappa)^2} = \frac{\kappa^2 \|w_0\|_2^2}{(1+\kappa)^2 - \phi},
\end{aligned}
$$

from which the result follows. $\qquad\square$

We now need to estimate $\delta^\top H \delta$ for a deterministic psd matrix $H$. Observe that

$$
\begin{aligned}
\delta^\top H \delta &= (Q_{n-1}w_0 - w_0)^\top H(Q_{n-1}w_0 - w_0) \\
&= w_0^\top Q_{n-1}^\top H Q_{n-1} w_0 - 2 w_0^\top Q_{n-1}^\top H w_0 + w_0^\top H w_0.
\end{aligned}
\tag{45}
$$

### F.7  Proof of Theorem 4.5 and Theorem 4.6 (Model Collapse in the Absence of Label Noise)

We first prove Theorem 4.6. Note that since we are in the isotropic case, $\Delta Bias$ defined in (15) is now given by $\Delta Bias := \mathbb{E} \|\widehat{\Sigma} R(Q_{n-1}w_0 - w_0)\|^2$, where $Q_{n-1} := P_{n-1}P_{n-1}\dots P_0$. Moreover, since $T > d$ and $\lambda = 0$ by assumption, we have $\Sigma R = I_d$, and so we further have $\Delta Bias := \mathbb{E}\|Q_{n-1}w_0 - w_0\|^2$. Now, one computes

$$
\begin{aligned}
\|Q_{n-1}w_0 - w_0\|^2 &= \|w_0\|^2 - 2 w_0^\top Q_{n-1}w_0 + w_0^\top Q_{n-1}^\top Q_{n-1}w_0 \\
&= \|w_0\|^2 - w_0^\top Q_{n-1}w_0 \\
&\simeq \|w_0\|^2 - w_0^\top \left( \prod_{m=0}^{n-1} (I + \kappa_m I)^{-1} \right) w_0 \\
&= \|w_0\|^2 - \|w_0\|^2 \prod_{m=0}^{n-1} \min(1/\phi_m, 1),
\end{aligned}
\tag{46}
$$

where on the 2nd line we have used the fact that $Q_{n-1}^\top Q_{n-1} = Q_{n-1}$ because the $P_m$'s are projections; on the 3rd line we have used Lemma F.7 with $\Sigma = I$ and $u = v = w_0$; on the 4th line

we have used the fact that $\kappa_m := \kappa(0, T_m) = \max(\phi_m - 1, 0) = \max(\phi_m, 1) - 1$. This completes the proof of Theorem 4.6.

The proof of Theorem 4.5 is completely analogous, with $Q_{n-1}$ replaced with $Q_0$. $\qquad\square$

**Lemma F.7.** *Let $X_0, \ldots, X_{n-1}$ be independent random matrices of shapes $T_m \times d$ for $m = 0, \ldots, n-1$, with rows iid from $N(0, \Sigma)$, and let $Q_{n-1} := P_{n-1}P_{n-2}\ldots P_0$, where $P_m = X_m^\dagger X_m$ is the orthogonal projection onto the subspace of $\mathbb{R}^d$ spanned by the rows of $X_m$. Then, in the limit $d, T_0, \ldots, T_{n-1} \to \infty$ such that $d/T_0 \to \phi_0 \in (0, \infty), \ldots, d/T_{n-1} \to \phi_{n-1} \in (0, \infty)$ with $\|\Sigma\|_{op}, \|\Sigma^{-1}\|_{op} = O(1)$, it holds that for deterministic $L_2$-bounded sequences of vectors $u$ and $v$*

$$u^\top Q_{n-1}v \simeq u^\top \left( \prod_{m=0}^{n-1} (\Sigma + \kappa_m I)^{-1} \right) v, \tag{47}$$

*where $\kappa_m = \kappa(0, T_m)$ is as defined in (9).*

*Proof.* The prof is by induction on $n \geq 1$. For $n = 1$, we have $Q_{n-1} = Q_0 = P_0$. Thus,

$$\begin{aligned} u^\top Q_0 v = u^\top P_0 v &= \lim_{t \to 0^+} u^\top X_0^\top (X_0 X_0^\top + tI)^{-1} X_0 v \\ &\simeq \lim_{t \to 0^+} u^\top (\Sigma + \kappa(t,T)I)^{-1} v = u^\top (\Sigma + \kappa_0 I)^{-1} v, \end{aligned} \tag{48}$$

where $\kappa_0 := \kappa(0, T)$ and we used Proposition 2 of Bach [4] at the beginning of the 2nd line. Now, suppose the claim holds for $n$, and let's prove that it holds for $n + 1$. Indeed,

$$u^\top Q_n v = u^\top P_n Q_{n-1} v \simeq u^\top P_{n-1} \prod_{m=0}^{n-1} (\Sigma + \kappa_m)^{-1} v \simeq u^\top \prod_{m=0}^{n} (\Sigma + \kappa_m)^{-1} v,$$

where the second step is an application of the induction hypothesis with $P_n u$ in place of $u$. $\qquad\square$

The following lemma can be used to compute $\|Q_{n-1}w_0 - w_0\|_\Sigma^2$ in the case of anisotropic $\Sigma$.

**Lemma F.8.** *Under the hypothesis of Lemma F.7, it holds that*

$$u^\top Q_{n-1} v \simeq u^\top \Sigma^n \left( \prod_{m=0}^{n-1} (\Sigma + \kappa_m I)^{-1} \right) v, \tag{49}$$

$$u^\top Q_{n-1}^\top \Sigma Q_{n-1} v \simeq u^\top \Sigma^n \left( \prod_{m=0}^{n-1} A_m \right) v, \tag{50}$$

$$\text{with } A_m := (\Sigma + \kappa_m I)^{-2} \left( \Sigma^2 + \frac{\kappa_m^2 \mathrm{df}_2(\kappa_m)}{T - \mathrm{df}_2(\kappa_m)} I \right), \tag{51}$$

*where $\kappa_m := \kappa(0, T_m)$ as defined in (9).*

*Proof.* The first formula follows directly from Lemma F.7 with $u$ replaced with $\Sigma u$. For the second formula, we can write

$$\begin{aligned} u^\top Q_{n-1}^\top M Q_{n-1} v &= u^\top P_0 P_1 \ldots P_{n-2} P_{n-1} M P_{n-1} P_{n-2} \ldots P_0 P_1 v \\ &= \widetilde{u}_{n-1}^\top P_{n-1} M P_{n-1} \widetilde{v}_{n-1}, \end{aligned}$$

where $\widetilde{u}_{n-1} := P_{n-2} \ldots P_0 u$. So we really only need to prove the result for $n = 1$; the general case will follow by induction and due to multiplicativity. Indeed, defining $A = \Sigma^{1/2} u v^\top \Sigma^{1/2}$, $B = \Sigma^{1/2} M \Sigma^{1/2}$, and $Z_0 = X_0 \Sigma^{-1/2}$, we have

$$\begin{aligned} u^\top P_0 M P_0 v &= \lim_{t \to 0^+} u^\top X_0^\top (X_0 X_0^\top + tI)^{-1} X_0 M X_0^\top (X_0 X_0^\top + tI)^{-1} X_0 v \\ &= \lim_{t \to 0^+} \mathrm{tr}\, A Z_0 (Z_0 \Sigma Z_0^\top + tI)^{-1} Z_0 B Z_0^\top (Z_0 Z_0^\top + tI)^{-1} Z_0 \\ &\simeq \mathrm{tr}\, A(\Sigma + \kappa_0 I)^{-1} B (\Sigma + \kappa_0 I)^{-1} + \kappa_0^2 \,\mathrm{tr}\, A(\Sigma + \kappa_0 I)^{-2} \cdot \frac{\mathrm{tr}\, B(\Sigma + \kappa_0 I)^{-2}}{T - \mathrm{df}_2(\kappa_0)} \\ &= u^\top (\Sigma + \kappa_0 I)^{-1} \Sigma M \Sigma (\Sigma + \kappa_0 I)^{-1} v + \kappa_0^2 u^\top \Sigma (\Sigma + \kappa_0 I)^{-2} v \cdot \frac{\mathrm{tr}\, M \Sigma (\Sigma + \kappa_0 I)^{-2}}{T - \mathrm{df}_2(\kappa_0)} \\ &= u^\top \Sigma A_0 v, \text{ for } M = \Sigma, \end{aligned}$$

where the 3rd line is an application of Proposition 2 of Bach [4]. $\qquad\square$

## G   The Noisy Regime for Power Law Spectra

Here we discuss the consequences of Theorem 5.1 for the noisy regime.

Now fix $\sigma^2 \neq 0$ and $\phi_0 \in (0,1)$. In this regime, Theorem 5.1 predicts that consistency (i.e. $E_{test}(\widehat{w}_n^{pred}) \overset{T \to \infty}{\longrightarrow} 0$) is only possible if $\ell \leq \ell_\star$. First consider values of $\ell$ for which the clean variance $\sigma^2 T^{-(1-\ell/\beta)}$ is less than the clean bias $T^{-2r\ell}$ in (27) i.e. $0 \leq \ell \leq \ell_{crit}$. We get

$$E_{test}(\widehat{w}_n^{pred}) \asymp T^{-2\ell \underline{r}} + T^{-(b-a-\ell/\beta)},$$

which is minimized by taking $\ell = \min(\ell_\star, \ell_{crit})$. For other values of $\ell$, variance dominates, giving

$$E_{test}(\widehat{w}_n^{pred}) \asymp T^{-(1-\ell/\beta)} + T^{-(b-\ell/\beta-a)} \asymp T^{-(\gamma-\ell/\beta)},$$

where $\gamma := \min(1, b-a)$. This is minimized by taking $\ell = \ell_{crit}$, leading to

$$E_{test}(\widehat{w}_n^{pred}) \asymp T^{-(\gamma-1/(2\beta\underline{r}+1))}.$$

This tends to zero with $T \to \infty$ only if $b > a + 1/(2\beta\underline{r} + 1)$.

## H   Proof of Results for Power-Law Covariance Spectrum

### H.1   Proof of Theorem 5.1

From Theorem F.1, we need to analyze the quantity

$$\rho \simeq \frac{\mathrm{df}_2(\kappa(\lambda))}{T_0 - d} + \frac{\kappa(\lambda)^2 \operatorname{tr}\left(\Sigma + \kappa(\lambda)I_d\right)^{-2}}{T_0 - d} \cdot \frac{\mathrm{df}_2(\kappa(\lambda))}{T - \mathrm{df}_2(\kappa(\lambda))}. \tag{52}$$

Now, for small $\lambda$, $\kappa := \kappa(\lambda)$ is small and one can compute

$$\mathrm{df}_m(\kappa) \asymp \sum_i \frac{\lambda_i^m}{(\lambda_i + \kappa)^m} = \kappa^{-m} \sum_i \frac{\lambda_i^m}{(1 + \kappa^{-1}\lambda_i)^m} \asymp \kappa^{-m}\kappa^{(m-1/\beta)} = \kappa^{-1/\beta},$$

where we have used Lemma I.1 with $D = \kappa^{-1}$ and $n = m$ in the last step. On the other hand, we can use some of the results of Appendix A (Section 3) of [14] to do the following. It can be shown (see aforementioned paper) that

- If $\ell > \beta$, then $\kappa \asymp T^{-\beta}$, and so $\mathrm{df}_m(\kappa) \asymp T$ for all $m \geq 1$.
- If $\ell < \beta$, then $\kappa \asymp \lambda \asymp T^{-\ell}$, and so $\mathrm{df}_m(\kappa) \asymp T^{\ell/\beta} = o(T)$ for all $m \geq 1$.

For $\ell < \beta$, plugging this into (52) gives

$$\begin{aligned}
\rho &\asymp \frac{T^{\ell/\beta}}{T_0 - d} + \frac{d}{T_0 - d}\frac{T^{\ell/\beta}}{T - T^{\ell/\beta}} \asymp T_0^{-1}T^{\ell/\beta} + \frac{\phi_0}{1 - \phi_0}T^{-(1-\ell/\beta)} \\
&\asymp \frac{1}{1 - \phi_0}\max\left(T/T_0, \phi_0\right)T^{-(1-\ell/\beta)},
\end{aligned}$$

where $\phi_0 := d/T_0$. Combining our Theorem F.1 with (57), we get the claimed result.   $\square$

### H.2   Representation of Clean Test Error

We make a small digression to present the following curiosity: with a slight leap of faith, the main results of [14] can be obtained in a few lines from the tools developed in [4], namely Proposition 4.2. This is significant, because the computations in [14] were done via methods of statistical physics (replica trick), while [4] is based on RMT.

Indeed, for regularization parameter $\lambda \asymp T^{-\ell}$ given in (25), we have $\kappa = \kappa(\lambda) \simeq \lambda$. Thus

$$\kappa \asymp T^{-\ell}, \quad \mathrm{df}_2(\kappa) \asymp \kappa^{-1/\beta} \asymp T^{\ell/\beta}. \tag{53}$$

Now, since $\lambda_i \asymp i^{-\beta}$ (capacity condition) and $(w_0^\top v_i)^2 = c_i^2 \asymp i^{-\delta}$ (source condition), we deduce

$$
\kappa^2 w_0^\top \Sigma (\Sigma + \kappa I)^{-2} w_0 \asymp w_0^\top \left( \sum_i \frac{\lambda_i}{(\lambda_i + \kappa^{-1}\lambda_i)^2} v_i v_i^\top \right) w_0 = \sum_i \frac{c_i^2 \lambda_i}{(\lambda_i + \kappa^{-1}\lambda_i)^2}
$$

$$
= \sum_i \frac{c_i^2 \lambda_i}{(\lambda_i + \kappa^{-1}\lambda_i)^2} \asymp \sum_i \frac{\lambda_i^{1+\delta/\beta}}{(\lambda_i + \kappa^{-1}\lambda_i)^2} \asymp \kappa^{-\gamma} \asymp T^{-\ell\gamma},
\tag{54}
$$

where $\gamma = \min(2, 1 + \delta/\beta - 1/\beta) = \min(2, 2r) = 2\underline{r}$, with $\underline{r} := \min(r, 1)$. The exponent is so because $\delta = 1 + \beta(2r - 1)$, and so $\delta/\beta = 1/\beta + 2r - 1$ by construction. The estimation of the last sum in (54) is thanks to Lemma I.1 applied with $D = \kappa^{-1}$, $n = 1 + \delta/\beta$, and $m = 2$. Therefore, invoking Proposition 4.2 gives

$$
Bias \simeq \frac{\kappa^2 w_0^\top \Sigma (\Sigma + \kappa)^{-2} w_0}{1 - \mathrm{df}_2(\kappa)/T} \asymp \frac{T^{-\ell\gamma}}{1 - T^{-(1-\ell/\beta)}} \asymp T^{-\ell\gamma} = T^{-2\ell\underline{r}}
\tag{55}
$$

$$
Var \simeq \sigma^2 \frac{\mathrm{df}_2(\kappa)}{T} \cdot \frac{1}{1 - \mathrm{df}_2(\kappa)/T} \asymp \sigma^2 \frac{T^{\ell/\beta}}{T} \frac{1}{1 - o(1)} \asymp \sigma^2 T^{-(1-\ell/\beta)}.
\tag{56}
$$

We deduce the scaling law

$$
E_{test} \simeq Bias + Var \asymp T^{-2\ell\underline{r}} + \sigma^2 T^{-(1-\ell/\beta)} \asymp \max(\sigma^2, T^{1-2\ell\underline{r}-\ell/\beta}) T^{-(1-\ell/\beta)},
\tag{57}
$$

which is precisely the main result of [14].

**Low-Noise Regime.** In the low noise regime where $\sigma^2 = O(T^{-2\beta\underline{r}})$, one may take $\ell = \beta$; the variance is then much smaller than the bias, and one has the fast rate

$$
E_{test} \asymp T^{-2\beta\underline{r}}.
\tag{58}
$$

**High-Noise Regime.** Now, consider the case where $\sigma^2 = \Theta(1)$. Setting $2\ell\underline{r} = 1 - \ell/\beta$ to balance out the bias and variance gives $\ell = \ell_{crit}$, where

$$
\ell_{crit} := \frac{\beta}{2\beta\underline{r} + 1} \in (0, \beta).
\tag{59}
$$

With this value of the exponent $\ell$, we get the error rate

$$
E_{test} \asymp T^{-2\ell_{crit} \cdot \underline{r}} = T^{-c}, \text{ with } c := \frac{2\beta\underline{r}}{2\beta\underline{r} + 1},
\tag{60}
$$

which is precisely the main result of [14], known to be minimax optimal (de Vito [11], etc.) !

# I   Auxiliary Lemmas

**Lemma I.1.** *Let the sequence $(\lambda_k)_{k\geq 1}$ of positive numbers be such that $\lambda_k \asymp k^{-\beta}$ for some constant $\beta > 0$, and let $m, n \geq 0$ with $n\beta > 1$. Then, for $D \gg 1$, it holds that*

$$
\sum_{k=1}^\infty \frac{\lambda_k^n}{(1 + D\lambda_k)^m} \asymp D^{-c} \begin{cases} \log D, & \text{if } m = n - 1/\beta, \\ 1, & \text{else,} \end{cases}
\tag{61}
$$

*where $c := \min(m, n - 1/\beta) \geq 0$.*

*Proof.* First observe that

$$
\lambda_k^n / (1 + D\lambda_k)^m \asymp \lambda_k^n \min(1, (D\lambda_k)^{-m})
$$

$$
= \begin{cases} \lambda_k^n = k^{-n\beta}, & \text{if } D\lambda_k < 1, \text{ i.e if } k > D^{1/\beta}, \\ D^{-m} \lambda_k^{-(m-n)} = D^{-m} k^{(m-n)\beta}, & \text{else.} \end{cases}
$$

We deduce that

$$
\sum_{k=1}^\infty \frac{\lambda_k^n}{(1 + D\lambda_k)^m} \asymp D^{-m} \sum_{1 \leq k \leq D^{1/\beta}} k^{(m-n)\beta} + \sum_{k > D^{1/\beta}} k^{-n\beta}.
\tag{62}
$$

By comparing with the corresponding integral, one can write the first sum in (62) as

$$D^{-m} \sum_{1 \le k \le D^{1/\beta}} k^{(m-n)\beta} \asymp D^{-m} \int_1^{D^{1/\beta}} u^{(m-n)\beta} \mathrm{d}u$$

$$\asymp D^{-m} \begin{cases} (D^{1/\beta})^{1+(m-n)\beta} = D^{-(n-1/\beta)}, & \text{if } n - 1/\beta < m, \\ \log D, & \text{if } m = n - 1/\beta, \\ 1, & \text{else.} \end{cases}$$

$$= \begin{cases} D^{-(n-1/\beta)}, & \text{if } n - 1/\beta < m, \\ D^{-m} \log D, & \text{if } m = n - 1/\beta, \\ D^{-m}, & \text{else.} \end{cases}$$

$$= D^{-c} \begin{cases} \log D, & \text{if } m = n - 1/\beta, \\ 1, & \text{else,} \end{cases}$$

where $c \ge 0$ is as given in the lemma.

Analogously, one can write the second sum in (62) as

$$\sum_{k > D^{1/\beta}} k^{-n\beta} \asymp \int_{D^{1/\beta}}^\infty u^{-n\beta} \mathrm{d}u \asymp (D^{1/\beta})^{1-n\beta} = D^{-(n-1/\beta)},$$

and the result follows upon putting things together. $\qquad\square$

**Lemma I.2.** *For $\kappa = \kappa(\lambda, T)$ defined as in (9), it holds that*

$$\frac{\partial \kappa}{\partial \lambda} = \frac{1}{1 - \mathrm{df}_2(\kappa)/T} \ge 1. \tag{63}$$

The formula given in the above lemma is useful because it can be combined with the identities

$$Bias = w_0^\top \Sigma (\Sigma + \kappa I)^{-2} w_0 \frac{\partial \kappa}{\partial \lambda}, \tag{64}$$

$$Var = \sigma^2 \frac{\mathrm{df}_2(\kappa)}{T} \frac{\partial \kappa}{\partial \lambda}. \tag{65}$$

The RHS of (64) is usually referred to as the omniscient risk Hastie et al. [22], Cui et al. [13], Wei et al. [48].

*Proof of Lemma I.2.* By definition of $\kappa$, we know that

$$\kappa - \lambda = \kappa \mathrm{df}_1(\kappa)/T = \kappa \operatorname{tr} \Sigma (\Sigma + \kappa I)^{-1}/T.$$

Let $\kappa := \frac{\partial \kappa}{\partial \lambda}$. Differentiating the above identity w.r.t. $\lambda$ gives

$$\kappa' - 1 = \kappa'(\operatorname{tr} \Sigma (\Sigma + \kappa I)^{-1} - \kappa \operatorname{tr} \Sigma (\Sigma + \kappa)^{-2})/T = \kappa' \operatorname{tr} \Sigma^2 (\Sigma + \kappa I)^{-2}/T = \kappa' \mathrm{df}_2(\kappa)/T,$$

and the result follows upon rearranging. Note that we have used the identity

$$I - \kappa (\Sigma + \kappa I)^{-1} = \Sigma (\Sigma + \kappa I)^{-1},$$

to rewrite $\Sigma (\Sigma + \kappa I)^{-1} - \kappa \Sigma (\Sigma + \kappa I)^{-2} = \Sigma^2 (\Sigma + \kappa I)^{-2}$. $\qquad\square$

