# OpenReview forum: "Model Collapse Demystified: The Case of Regression"
_NeurIPS.cc/2024/Conference — NeurIPS 2024 poster_

### Official Review · Reviewer_jLLV · 2024-07-11

**Soundness:** 3
**Presentation:** 3
**Contribution:** 3
**Rating:** 7
**Confidence:** 3

**Summary:**

This paper analyses a ridge regression model of the recently described phenomenon of model collapse in various settings (unregularised linear regression, large dimensional RMT limit regression), acquiring analytical expressions for scaling limits in the case of heavy tails under capacity and source conditions, which are validated against numerical experiments.

**Strengths:**

The model presented in the pair is well-presented, and put into context in terms of extra analytical term additions that arise due to recursive re-training on synthetic data. This is particularly interesting for analysing the phenomenon of model collapse, as it highlights that scaling laws must be adjusted when synthetic data appears in the training data of the model, and further highlights the necessity of adjustable regularization - directly deriving how the optimal regularisation choice changes due to synthetic data. This places the paper in a very good spot as a first step towards understanding how to mitigate model collapse.

**Weaknesses:**

There are a number of weakness that this paper does not address in extensive detail. To begin with, the model considered here is extremely simplistic and places itself much closer to the distillation literature, rather than generative literature. I believe too much emphasis is made on the linear regression, which does not seem to provide significantly more insights beyond that provided in the original model collapse paper by Shumailov et al. There may be more, which I may not be seeing, but they dont appear to be mentioned in the paper explicitly. The particularly interesting section establishing the scaling laws, which goes significantly beyond the model in Shumailov et al. seems to be only briefly discussed, and the novel insights from this analysis are not discussed beyond the paragraph in the introduction.

While it may be interesting as a generic model, the potential insights it might bring into the process of model collapse are much more important, and for this it needs to be put into context of the limitations of previous proposed models of the phenomenon.

In addition, the paper multiple times claims to analyse the kernel ridge regression model. However, *none* of the models described are in the kernel regression setting. While it may be an uncomplicated extension, as discussed in Appendix B, the current version of the paper does not perform this extension - and none of the results are presented in this setting. Thus, currently the paper is overselling its contributions.

**Questions:**

The connection to self-distillation work is briefly mentioned the appendix, however it is mentioned that it stands apart from it. I am not clear how this is the case? This is especially not true in the noiseless label setting, when design matrices are fixed to be the same. When either of those isnt true, we still find ourselves in the distillation setting, alas not exactly the model considered in Mobahi et al. Could you clarify this either for me or in the paper?

Section 4 ends very abruptly, and no in-depth discussion of the model is presented. I believe this to be a significant limitation, as relevance of the model becomes unclear.


Further comments:
* line 30 - kernel regression is not analysed in this paper
* line 47 - you do not mention what \epsilon is
* line 54 - Insights from this formula needs to be related to existing mathematical models of model collapse.
* line 118 - \hat{\omega} is a vector input to E_{test}? Or is it a random variable for expectation?
* It does not seem to be clarified what the effect of sampling different design matrices is.
* around line 128 - notation seems a bit overloaded, with n subscripts being dropped, even though you may not have the exact same variables across iterations? E.g. X_n can not always be denoted by X?
* line 155 - the whole paragraph seems to be describing almost the same result as Shumailov et al., but does not mention it.
* line 173 missing citation
* I am slightly unclear why section 3.4 is underparametrized - could you clarify?

**Limitations:**

See above responses

---

> ### Author Rebuttal · Authors · 2024-08-07
>
> Thank you for your thoughtful review and the dedicated time and effort you invested. We genuinely appreciate your recognition of the key strengths in our work.
>
> 1. > To begin with, the model considered here is extremely simplistic and places itself much closer to the distillation literature, rather than generative literature ...
>
> Thanks for the question. We have addressed it in the general response, and will include a discussion comparing our results with previous theoretical results in the paper.
>
> 2. > The particularly interesting section establishing the scaling laws, which goes significantly beyond the model in Shumailov et al. seems to be only briefly discussed, and the novel insights from this analysis are not discussed beyond the paragraph in the introduction.
>
> The purpose of Section 4 to which the reviewer is referring is to quantifiable show how model collapse manifests as a **change of scaling laws**, in a setting which is close to neural networks (in certain regimes) and large language models. We recall that scaling laws (Kaplan et al. [26] Hoffmann et al. [24]), which relate the test error of a model to the sample size, model size, and amount of compute used, are one of the main instruments used by practitioners in strategic allocation of resources in the design and implementation of large language models (ChatGPT, Llama, etc.). Indeed, the results of Section 4 predict that the scaling laws w.r.t dataset slow down (i.e smaller exponents) in the presence of **synthetic data**: a much larger sample size might be required to achieve the same decrease in test error as with real data. Moreover, the test error for models trained on **synthetic data** will eventually plateau as a function of sample size, in sharp contrast to the idealized classical picture [24, 26] where the test error for models trained on **real data** is predicted to decrease forever. These theoretical results are illustrated in Figure 1(b) and Figure 4.
>
> We will add the discussion to the paper.
>
> 3. > While it may be interesting as a generic model, the potential insights it might bring into the process of model collapse are much more important, and for this it needs to be put into context of the limitations of previous proposed models of the phenomenon.
>
> We have addressed the point raise here in the general response and will include it in the paper.
>
> 4. > The kernel ridge regression model.
>
> Thank you for the question. The main difference between linear regression and kernel regression lies in the use of the kernel feature mapping. In our setting, generalizing from linear to kernel regression is straightforward. We chose to use linear regression because it is simpler in terms of both notation and understanding. As noted at line 47, "we present the linear regression setting for simplicity; the generalization to the kernel setting is straightforward." and we have discussed it in more details in Appendix B. We will also add a footnote at line 30.
>
> 5. > The connection to self-distillation.
>
> Thank you for raising this point, which caused us some grief to formulate succintly in the paper. Allow us to expand here, to address your very important question (Model collapse versus self-distillation):
> Our setting is inspired by the model collapse phenomenon, where more and more there will be vast amounts of synthetic data generated by users posted online, which will necessarily enter the training set of the next foundation model. In this case, we do not have the groud truth label, and the generation of synthetic data is not controlled by us, but by other users. Therefore, we adopt the setting with only synthetic labels with added noise.
>
> Specifically, in our setup, at generation $n>0$, one doesn't have access to the true labels $Y_0=f_0(X) + noise$ for the training samples $X$, but rather to some $\hat Y_n = \hat f_n(X) + noise$, where $\hat f_n$ is an unknown function, which synthesizes fake labels iteratively; the integer $n$ is the number of iterations. $f_0$ is the ground-truth labelling function (unknown). In our work, we make the structural assumption that $\hat f_n$ is obtained by iterative / successive regressions on a true dataset $D_0=(X_0,Y_0)$, and then new noise is injected every single generation. The fake labels $\hat Y_n$ are like labels for images $X$ on internet, or outputs from a LLM like ChatGPT. One doesn't have any control over the creation of these labels, especially because of the noise injected at each stage. In practice, one doesn't even know that they're not real labels. The final model is learnt on the dataset $(X,\widehat Y_n)$.
>
> In the **self-distillation** setting, one has access to the true labels $Y$. One decides to replace $Y$ with some $Y_n := F_n(X,Y)$. Here, $f_0$ is the ground-truth labelling function (i.e $f_0(x) := x^\top w_0$ in the case of linear models). The final model learned is via regression on the dataset $(X,Y_n)$. It is up to the practitioner to chose what the number of iterations $n$, and in fact Mobahi et al. showed that there is an optimal value of $n$ depending on the distribution of the data and the model class over which the regression is done. This is what we mean by having control over the data synthesization process. Thus, mathematically, the difference boils down to the absence of fresh noise when data is regenerated in SD, which leads to completely different behavior to our setting, where we do not control the noise injected at each generation. **Allow us to use a metaphore here. Model collapse is like the rain, while SD is like the shower. The latter can be controlled by the user (for example, turned on / off), while the user has to cope with the former.**
>
> 6. > Section 4 ends very abruptly, and no in-depth discussion of the model is presented.
>
> Thank you for the question. We agree that additional explanation is necessary. We have provided a discussion in response to question 2 and will include it in the paper.

---

> > ### Comment · Reviewer_jLLV · 2024-08-09
> >
> > I would like to thank the authors for their answers and clarifications. I have a few comments for the clarifications provided.
> >
> > * As noted at line 47, "we present the linear regression setting for simplicity; the generalization to the kernel setting is straightforward." and we have discussed it in more details in Appendix B.
> >
> > This was clear from the paper and the concern is not with that statement, but rather with the claim in the introduction and a few other places, that in fact your paper does analyse the kernel ridge regression model. This is *not* the case and should not be claimed. I believe it is fine to mention that the extension may be straightforward, but it is *wrong* to claim that your perform this analysis.
> >
> > * The connection to self-distillation.
> >
> > I do not believe there is any doubt that self-distillation is a different process from model collapse. However the two are related, and in fact your *theoretical* model falls closer to *theoretical* models of self-distillation, due to the regressive nature of your model. In fact, in the noiseless case the theoretical model is *exactly* the model of Mobahi et al, even though the aim of your model is different. The noisy case *does* place itself apart from the model of Mobahi et al, but can easily be viewed as a noisy version of self-distillation.
> >
> > I once again emphasise, it is not the processes that are similar. It is the theoretical models.
> >
> > "and in fact Mobahi et al. showed that there is an optimal value of depending on the distribution of the data and the model class over which the regression is done. " - I am not sure we are reading the same paper, which result are you referring to?

---

> ### Author Response · Authors · 2024-08-07
> **On further comments**
>
> **Please allow us to use the official comment section to address the further comments you have in your review.**
>
> 7. > line 30 - kernel regression is not analysed in this paper
>
> We have addressed the point raised here in the response to question 4.
>
> 8. > line 47 - you do not mention what \epsilon is
>
> Thanks. $\epsilon$ is the label noise, and we will add it.
>
> 9. > line 54 - Insights from this formula needs to be related to existing mathematical models of model collapse.
>
> We have addressed this point in the general response by comparing our theory with previous ones. We will also connect it with the results from Shumailov et al., as their work is the most similar to ours in mathematical form.
>
> 10. > line 118 - $\hat{\omega}$ is a vector input to $E_{test}$? Or is it a random variable for expectation?
>
> Sorry for the confusion. $\hat{w}$ is the input and there is no expectation over it. The notation $E_{test}(\hat w)$ stands for test error of the downstream model $\hat w$ as defined in equation (3).
>
> 11. > It does not seem to be clarified what the effect of sampling different design matrices is.
>
> In the "Dependent Case" at line 247, we analyze the scenario where the same design matrix is used for all generations, i.e $X_m=X_0$ for all $m<n$. Further restricting to the case $T_m=T_0 < d$ for all $m<n$ (i.e over-parametrized generator), our Theorem 3.5 shows that in this case, even in the absence of any label noise, there is model collapse due to an increase in the bias term in the test error. This increase in proportional to $1-\eta_0$, where $\eta_0:=T_0/d<1$. In the "Independent Case" on line 257, we study the scenario where an independent design matrix $X_m$ with $T_m<d$ rows is resampled at each stage $m<n$ of the generation process. In this case, our Theorem 3.6 shows that increase in bias is proportional to $1-\eta_0\eta_1\ldots\eta_{n-1}$, which is typically much larger than $1-\eta_0$.
>
> 12. > around line 128 - notation seems a bit overloaded, with n subscripts being dropped, even though you may not have the exact same variables across iterations? E.g. X_n can not always be denoted by X?
>
> In our theory, we focus solely on the last generation, as illustrated by the purple boxes in Figure 3. Since we fix all previous generations and examine only the last one, it is consistent to denote $X_n$ with $X$.
>
> 13. > line 155 - the whole paragraph seems to be describing almost the same result as Shumailov et al., but does not mention it.
>
> We will acknowledge Shumailov et al. as we arrive at a similar expression depending on the number of samples across all generations. However, our discussion focuses on how to analytically mitigate model collapse by using more samples during generation and whether this is an effective method.
>
> 14. > line 173 missing citation
>
> It is intended to be a question mark. We have removed the question mark for better understanding.
>
> 15. > I am slightly unclear why section 3.4 is underparametrized - could you clarify?
>
> As stated in the beginning the section, this refers to the scenario where each stage of the recursive synthetic data generator is based on at most as many samples as input dimensions, i.e $T_0 \ge d$. In this scenario, there are more data points than dimensions, resulting in a unique solution for the regression. Our theory reveals that in this setup, model collapse is due to an increase in the variance of the downstream estimator. Conversely, when $T_0 < d$, we refer to the synthetic data generator as over-parameterized, as there is no unique solution to the regression problem without regularization. In such cases, various interpolators can achieve training error:  OLS produces the one with minmal Euclidean norm. As we see in Section 3.5, model collapse appears in this scenario even in the abscence of label noise
>
> We thank the reviewers for all their questions and hope we have addressed them thoroughly.

---

> ### Author Response · Authors · 2024-08-09
>
> We thank the reviewer for their further comments.
>
> - About kernel regression
>
> We agree with the reviewer that the statement about kernel regression should be a remark that our analysis can be straightforwardly extended to the the case of kernel regression. This will be rectified.
>
> -  About self-distillation
>
> We would like to stress that our setup **cannot** be reduced to a **noisy version of self-distillation**. Let us explain. Self-distillation for linear regression would amount to a very special **instance** of our analysis where (1) $X_0=X_1=\ldots=X_{n-1}=X_n=X$ and (2) $\sigma_0=\ldots=\sigma_{n-1}=0$. That is, there is exactly one design matrix which is used in the data generation process and in the downstream estimator, and also no additional source of label noise is present at the end of each generation.
>
> In the general setup considered in our work, (1) is not imposed. We typically assume that $X_0,X_1,\ldots,X_{n-1},X_n$ with $X_n=X$, are all independent random matrices. An exception is line 247 ("The Dependent Case") of Section 3.5, where we assume $X_m=X_0$ for all $m \le n-1$, and independent of $X_n=X$. That setup (considered for the purposes of showing that model collapse can still occur in the absence of label noise) also assumes $\sigma_m=0$ for all $m$; the analytic picture which emerges (Theorem 3.5) is already drastically different from what one would get from self-distillation (corresponding to additional assumption that $X=X_0$).
>
> Thus, not only are the processes in self-distillation and model collapse different, the theoretical models are drastically different.
>
> We will add this discussion in the main paper.
>
> >I am not sure we are reading the same paper, which result are you referring to?
>
>
> By "an optimal value of $n$" we refer to the number of distillation steps that yields the best performance, as discussed in Mobahi et al. (ref [36] of our manuscript). Specifically, the authors boldface highlight it in the introduction (specifically, the contributions paragraph) like so "This implies that a few rounds of self-distillation may reduce over-fitting, further rounds may lead to under-fitting and thus worse performance". This is accordance with the theory developed in their paper, and is empirically confirmed in their Figure 3.
>
> [36] Hossein Mobahi et al. "Self-distillation amplifies regularization in hilbert space." Advances in Neural Information Processing Systems 33 (2020)

---

> > ### Comment · Reviewer_jLLV · 2024-08-11
> >
> > I would like to thank the authors for engaging and clarfications. I will shortly raise the score.
> >
> > One quick unrelated comment - I am not sure that Mobahi et al. actually "showed that there is an optimal value of depending on the distribution of the data and the model class over which the regression is done." In the sense that this is the intuitive explanation, yes, and it is in line with the theory, but I dont believe it directly leads from the theory presented there.

---

> > > ### Author Response · Authors · 2024-08-12
> > >
> > > Thank you for appreciating our responses!
> > >
> > > We acknowledge your comment. Yes, the statement quoted are indeed not theoretically proven. They are supported by empirical evidence and are in line with theoretical insights.

---

### Official Review · Reviewer_Z9jq · 2024-07-12

**Soundness:** 3
**Presentation:** 3
**Contribution:** 2
**Rating:** 5
**Confidence:** 3

**Summary:**

This paper provides a theoretical analysis of the "model collapse" phenomenon that can occur when machine learning models are trained on synthetic data generated by other AI models. The key contributions include:

1. A mathematical framework for studying model collapse in high-dimensional regression settings.

2. Exact analytic formulae for how test error increases as a function of the number of synthetic data generations, regularization, sample sizes, and other parameters.

3. Demonstration of how model collapse can lead to a change in scaling laws, with test error potentially plateauing or even diverging rather than continuing to decrease with more data.

4. Analysis showing that even in noiseless settings, overparameterized synthetic data generators can lead to catastrophic model collapse.

5. Insights on how to potentially mitigate model collapse through adaptive regularization.

The authors provide theoretical results as well as empirical validation through experiments. Overall, the paper offers a rigorous mathematical treatment of model collapse, providing insights into how and why performance degrades when training on iteratively generated synthetic data.

**Strengths:**

1. The paper provides a solid mathematical framework for analyzing model collapse, offering precise analytical formulae rather than just empirical observations.

2. It covers various scenarios, including different dimensionality regimes, noise levels, and spectral conditions, providing a broad understanding of the phenomenon.

3. The work reveals new phenomena, such as the change in scaling laws and the potential for catastrophic collapse even in noiseless settings, advancing our understanding of model behavior with synthetic data.

4. The analysis of regularization effects and the proposal for adaptive regularization offer actionable insights for practitioners dealing with synthetic data.

5. While primarily theoretical, the paper validates its findings with experiments, strengthening the connection between theory and practical applications.

**Weaknesses:**

1. The analysis primarily focuses on regression tasks, which may not fully capture the complexities of more advanced machine learning models like large language models or image generators.

2. The theoretical framework relies on certain simplifying assumptions (e.g., Gaussian data distribution, linear models) that may not always hold in real-world scenarios.

3. While the paper includes experiments, they are primarily on synthetic datasets. Validation on real-world, large-scale datasets could further strengthen the findings.

4. While the paper proposes adaptive regularization as a potential mitigation strategy, it doesn't extensively explore or compare other possible solutions to model collapse.

**Questions:**

1. While you propose adaptive regularization as a potential solution to mitigate model collapse, could you elaborate on other practical strategies that might be effective? For instance, how might techniques like data filtering, model ensembling, or continual learning potentially address the issues you've identified?

2. Given your findings on how model performance can degrade with synthetic data, what implications do you see for AI safety and robustness, especially as AI-generated content becomes more prevalent on the internet? How might your work inform strategies for maintaining model quality and reliability in an ecosystem increasingly populated by AI-generated data?

**Limitations:**

1. The analysis primarily focuses on linear regression models, which may not fully capture the complexities of advanced machine learning architectures like deep neural networks or transformers used in modern AI systems.

2. The theoretical framework relies on idealized data distributions (e.g., Gaussian) and noise models, which may not accurately represent the diverse and complex nature of real-world datasets used in training large AI models.

3. While the paper includes experimental validation, it primarily uses synthetic datasets. The absence of experiments on large-scale, real-world datasets limits the immediate applicability of the findings to practical scenarios.

---

> ### Author Rebuttal · Authors · 2024-08-07
>
> We appreciate your review and positive evaluation of our work, as well as your acknowledgment of its strengths.
>
> 1. > The analysis primarily focuses on regression tasks, which may not fully capture the complexities of more advanced machine learning models like large language models or image generators. The theoretical framework relies on certain simplifying assumptions (e.g., Gaussian data distribution, linear models) that may not always hold in real-world scenarios. While the paper includes experiments, they are primarily on synthetic datasets. Validation on real-world, large-scale datasets could further strengthen the findings.
>
> We addressed it in the general response.
>
> 2. > While you propose adaptive regularization as a potential solution to mitigate model collapse, could you elaborate on other practical strategies that might be effective? For instance, how might techniques like data filtering, model ensembling, or continual learning potentially address the issues you've identified?
>
> Scaling laws tend to deteriorate when incorporating synthetic data. As demonstrated in our Theorem 4.1, adding more synthetic data can either break scaling laws or significantly worsen them. To achieve improvements in the next generation of foundational models, it is crucial to preserve these scaling laws. Data filtering is certainly beneficial in this regard. Additionally, watermarking synthetic data is becoming increasingly important as it aids in distinguishing high-quality real data from synthetic counterparts.
>
> Adaptive regularization in Corollary 4.2, akin to early stopping, suggests that early intervention during training can prevent model collapse. Furthermore, using the current model to filter and bootstrap the data could be a viable solution. This approach can help maintain the integrity of the training process by ensuring that only the most relevant and high-quality data, whether real or synthetic, is used.
>
> We will add the discussion to the paper.
>
> 3. > Given your findings on how model performance can degrade with synthetic data, what implications do you see for AI safety and robustness, especially as AI-generated content becomes more prevalent on the internet? How might your work inform strategies for maintaining model quality and reliability in an ecosystem increasingly populated by AI-generated data?
>
> The key takeaway is that scaling alone is insufficient when synthetic data is involved; these data can alter traditional training paradigms, particularly as optimal regularizations change. Consequently, new algorithms with appropriate regularizations are necessary. Maintaining the quality and reliability of models requires rigorous validation processes focused on data quality. More effort must be directed toward data curation to ensure the integrity of training sets. Additionally, developing tools and methodologies for detecting AI-generated content, such as watermarking, is essential to distinguish synthetic data from real data and to maintain model reliability.
>
> We remain available to address any further questions you may have. We hope our responses address your questions and encourage you to consider raising your evaluation score. Thank you.

---

> > ### Comment · Reviewer_Z9jq · 2024-08-10
> >
> > Thanks for the response. I decide to keep my original score.

---

> > > ### Author Response · Authors · 2024-08-13
> > > **Results of Additional Experiments (Non-Gaussian Data)**
> > >
> > > Dear Reviewer,
> > >
> > > Thanks again for your feedback on our work. We would like to kindly inform you that in addition to our previous response to your review, we have added additional experiments during our discussion with Reviewer 16H4. These new experiments go beyond Gaussian data. We demonstrate that the results still hold with real data and a neural networks in a variety of relevant regimes included (linearized regimes like RF / NTK, and also fully-trained networks). The insights from theory on linear models continue to apply in realistic settings non-linear non-Gaussian settings. The results are at [link](https://openreview.net/forum?id=bioHNTRnQk&noteId=YEMds4fAIE), and will be included in our manuscript. These additional results show that the linear trend in model-collapse (the increase in test error as a function of number of generations $n$ in the synthetic process) is likely be worst (superlinear / quadratic) in the case of large fully-trained finite-width neural networks.
> > >
> > > Reviewer 16H4 has also expressed strong support for our response, especially the aforementioned further experiments. We hope these discussions further address your concerns regarding weaknesses 1, 2, and 3, particularly the gap between the simplified setting and real-world scenarios.
> > >
> > > Thanks again,
> > > The Authors.

---

### Official Review · Reviewer_Xs1n · 2024-07-16

**Soundness:** 3
**Presentation:** 1
**Contribution:** 3
**Rating:** 5
**Confidence:** 3

**Summary:**

Authors analyze the problem of ´model collapse´, i.e., what happens when a generative model is retrained on data generated by itself. The authors study a slightly different and simplified setup: linear regression, where the target is iteratively generated by previous estimates of the coefficients. Based on random matrix theory tools authors provide sample complexity bounds on the estimation of the oracle regression coefficients.

**Strengths:**

The paper tackles an interesting and timely topic, the theoretical results seem pretty sophisticated and sharp

**Weaknesses:**

It seems that the provided analyses, in some kind of semi-supervised learning, do not match the motivations.
The motivation is clear, but the paper is overall hard to follow and often consists of a succession of arid theoretical results.
Either my area of expertise is too far, or the paper needs to be rewritten.

**Questions:**

- What´s the point of figure 1a? Illustrating the degradation of the models as they are retrained on synthetic data? What is the difference betzeen the solid and the dashed curves?
- What is $T$ exactly? Is this the number of synthetic samples at each step? The size of the initial dataset? Appendix did not provide clarifications to me on this question. This information seems vital to understand Figures 1a and 1b.
- I have a clarification question on the the theoretical setup: is a `semi-supervised` setup, where you assume that you have access to $Y_0, X_1, \dots, X_n$?

- On Theorem 3 (Equation 15), it seems that there is a remaining error $n \\sigma\_0 \\rho$, even if the model is perfect at step $0$. In other words, is it normal that if the model is perfect at step $0$, then the test error still increase with the number of retraining $n$?

---

> ### Author Rebuttal · Authors · 2024-08-07
>
> Thank you for your feedback, especially on the presentation of our material, which is challenging, given the amount of technical work required for our ((necessarily lengthy) proofs, and the difficulty of partitioning this content petween the main body and the appendix. Our aim for the main body of the paper was to present and to explain and connect the various technical results which are very hard to state in a simplified fashion. We take your point and will add a content overview style paragraph/picture to the paper (or the appendix if space does not permit this).
>
> 1. > What´s the point of figure 1a? Illustrating the degradation of the models as they are retrained on synthetic data? What is the difference betzeen the solid and the dashed curves?
>
> The figure illustrates three key aspects: the degradation with respect to the number of generations, the effect of regularization on accuracy, and the scaling law curve concerning the amount of data. The dashed curves represent theoretical predictions, while the solid curves depict experimental results.
>
> 2. > What is $T$ exactly?
>
> $T$ represents the number of samples generated by the generator to train the subsequent model, while $T_0$ denotes the number of samples used to train the previous generators. The complete definition can be found in Equation (6) and please see Figure 3 for an illustration of all our notation. We will enhance the caption of Figure 1 - thank you for pointing out the possible unclarity.
>
> 3. > I have a clarification question on the the theoretical setup: is it a semi-supervised setup, where you assume that you have access to $Y_0, X_1, X_2, ..., X_n$?
>
> No. We consider the fully supervised setting. We use $X_i$ to denote the input data at iteration $i$, and we only have $\bar Y_i, X_i$. In the paper, we define the data and labels used to train the subsequent models in Equation (5) within the box, and we further elaborate on this in Figure 3.
>
> 4. > On Theorem 3 (Equation 15), it seems that there is a remaining error $n \sigma_0 \rho$, even if the model is perfect at step 0. In other words, is it normal that if the model is perfect at step 0, then the test error still increase with the number of retraining?
>
> Yes. Consider the groud truth data as generated from a perfect model, and our theory still apply. The test error of consequent models still increase since there are finite sample bias and label noises.
>
> We thank you again for the appreciation of our work and are at your disposal if there are more questions. We hope our replies find your approval and would incite you to raise your score. Thank you.

---

> > ### Author Response · Authors · 2024-08-13
> > **End of Discussion Period**
> >
> > Dear Reviewer,
> >
> > Thanks again for your feedback on our work. Since we are getting close to the end of the discussion period, we would sincerely appreciate your response to our rebuttal. We would like to highlight that we have added additional experiments during the discussion, which you can find at [link](https://openreview.net/forum?id=bioHNTRnQk&noteId=YEMds4fAIE), and have addressed your concerns regarding the theoretical settings. Please let us know if we have adequately addressed your concerns, and we would be grateful if you would consider raising your score.
> >
> > Thanks in advance,
> >
> > The Authors

---

### Official Review · Reviewer_16H4 · 2024-07-20

**Soundness:** 4
**Presentation:** 2
**Contribution:** 3
**Rating:** 7
**Confidence:** 4

**Summary:**

This paper studies the model collapse problem in the context of ridge regression. The authors train a sequence of linear models which are trained on Gaussian data and characterize the test error under various settings. Their results are intuitive and theoretically support the ongoing research area of model collapse in generative models. For example, they show that the test error increases more for larger number of generations n in the ordinary least squares settings. They heavily use random matrix theory to derive their results.

**Strengths:**

1. The theoretical setup is sound and a good way to incorporate supervision in a generative model context in a way that is fitting for theoretical study.
2. The results are interesting and intuitive. In fact they reflect what we have seen empirically in several of the previous works.
3. Interestingly enough, since you use a supervised version of the model collapse, the martingale principle used in [2] does not apply. This makes this setting very interesting to study and I suggest that you mention this in your paper as it is a contribution which I don’t think that you claim.

**Weaknesses:**

Here are some things to be improved on in decreasing order of importance:
1. The main weakness of the paper is that it seems only tangentially related to the generative model collapse problem. This is for several reasons but most importantly because the kind of experimental and theoretical setup explained in this paper is not a complex generative model. The model collapse problem occurs with generative models which are quite complex (not linear models on gaussian data). Additionally, the results in the paper are all asymptotic, which don’t tell us too much about non-asymptotic behavior which we see in practice. That being said, the work done in this paper is actually really good and definitely is really polished and should be published. Therefore, experiments using deep learning models supporting the theoretical thesis of this paper are very important for this work to appear applicable to the generative model collapse problem.
2. The abstract is very vague and doesn’t describe what actually happens in the paper. Here are several areas of improvement:
- It is not mentioned in the abstract that the paper seems to focus primarily on Gaussian data, an important point.
- Moreover, “... quantitatively outline this phenomenon in a broad range of regimes.” Please say what regimes you are talking about otherwise this statement doesn’t really tell the reader almost any information.
- You mention “how model collapse affects both bias and variance terms” which is a very general statement but does this happen no matter what the model is? Or only with your specific setup? The evidence in the paper points to the latter and thus should be mention so that a reader does not think that this is a very general result.
- Results are pretty much all asymptotic. This is kind of hinted at by the mentioning of “scaling laws” but needs to be more explicit in order to reflect the work of the paper.
3. It is not clear what was mystified and now has been cleared up…. This is an important point because it is literally in the title of the paper.
4. Formatting:
- The axes in Figure 1 and 2 are way too small and hard to see.
- The flowchart in Figure 3 has tiny font too.
- You don’t have to change these, but I recommend not starting sentences with citations as in lines 35, 94, 98.
- On line 213 you mention that the formula for rho is in the appendix. I would not do this because then the reader has to go to the appendix to understand what rho is. Especially since you actually state what rho is in a special case on Equation (17) after redirecting the reader to the appendix.
- The conditions in Equation (22) need to be defined
5. Typos and mistakes:
- It seems that between equation (4) and (5) that there is a mistake in defining \widehat w_0
- For large d but actually for d->inf
- There is supposed to be no space on the section title on line 173

**Questions:**

None

**Limitations:**

The work is mostly on linear models with Gaussian data.

---

> ### Author Rebuttal · Authors · 2024-08-07
>
> We thank you for your thoughtful comments, and are delighted you appreciate the technical heavy lift that was required to prove our results and paint a picture of the regimes of "model collapse" in the setting of ridge regression. Thank you for pointing out that we should highlight what other technical arguments would fail (like martingale arguments) - we are adding this point to the next version!
>
>
> 1. > The main weakness of the paper is that it seems only tangentially related to the generative model collapse problem ... Therefore, experiments using deep learning models supporting the theoretical thesis of this paper are very important for this work to appear applicable to the generative model collapse problem.
>
> We addressed it in the general response.
>
> 2. > The abstract is very vague and doesn’t describe what actually happens in the paper.
> > It is not mentioned in the abstract that the paper seems to focus primarily on Gaussian data, an important point.
> Moreover, “... quantitatively outline this phenomenon in a broad range of regimes.” Please say what regimes you are talking about otherwise this statement doesn’t really tell the reader almost any information.
> You mention “how model collapse affects both bias and variance terms” which is a very general statement but does this happen no matter what the model is? Or only with your specific setup? The evidence in the paper points to the latter and thus should be mention so that a reader does not think that this is a very general result.
>
> Thank you for the suggestion! We have modified the abstract accordingly, with the differences marked in **bold**.
>
> The era of proliferation of large language and image generation models begs the question of what happens if models are trained on the synthesized outputs of other models. The phenomenon of "model collapse" refers to the situation whereby as a model is trained recursively on data generated from previous generations of itself over time, its performance degrades until the model eventually becomes completely useless, i.e. the model collapses. In this work, we investigate this phenomenon within the context of high-dimensional regression **with Gaussian data**, considering both low- and high-dimensional asymptotics. We derive analytical formulas that quantitatively describe this phenomenon in **both under-parameterized and over-parameterized regimes**. We show how test error increases linearly in the number of model iterations in terms of all problem hyperparameters (covariance spectrum, regularization, label noise level, dataset size) and further isolate how model collapse affects both bias and variance terms **in our setup**. We show that even in the noise-free case, catastrophic (exponentially fast) model-collapse can happen in the over-parametrized regime. In the special case of polynomial decaying spectral and source conditions, we obtain modified scaling laws which exhibit new crossover phenomena from fast to slow rates. We also propose a simple strategy based on adaptive regularization to mitigate model collapse. Our theoretical results are validated with experiments.
>
> > Results are pretty much all asymptotic. This is kind of hinted at by the mentioning of “scaling laws” but needs to be more explicit in order to reflect the work of the paper.
>
> We mentioned that "In this work, we study this phenomenon ..., under low- and high-dimensional asymptotics".
>
> 3. > It is not clear what was mystified and now has been cleared up…
>
> We have discussed how our theory compares with previous theories in the general response. Generally speaking, the concept of model collapse, as popularized by prior works, still lacks a theoretical framework to support empirically-backed findings. Our work addresses this gap by developing an analytic theory that clearly demonstrates how model collapse quantifiably emerges from training on synthetic data. This is reflected in the 'demystified' aspect of the title of our paper. We will incorporate this comparison into the paper.
>
> 4. > Formatting
>
> Thanks for the suggestion. We will change the font size.
>
> 5. > Typos and mistakes:
>
> Thanks a lot! We have corrected them.
>
> We would like to thank you once again for reviewing our work and helping us improve its presentation, especially with regard to weakness No.2. Please let us know if you have any more questions. If there are no outstanding concerns, we would kindly ask you to consider raising your score which would substantially help in reaching a reviewer consensus. Thank you.

---

> > ### Comment · Reviewer_16H4 · 2024-08-09
> > **Response to the rebuttal**
> >
> > I thank the authors for the rebuttal and for addressing most of my concerns! I have a few more questions regarding the paper and the rebuttal. I think that most of the comments were addressed except for two things:
> > 1. It still seems to me like the "mystified" part of the paper is a little bit ambigous. Specifically, the authors responded that "the concept of model collapse, as popularized by prior works, still lacks a theoretical framework to support empirically-backed findings" but also said that "In the area of model collapse, there exists a substantial body of literature [3, 4, 5], that provides theoretical analyses using Gaussian data". So these two statements seem contradicting. Being more specific about EXACTLY what is being demystified is important to make sure the work is well represented by the title.
> > 2. The authors mention that Gaussian data is used in other papers, but this paper is SOLELY regarding Gaussian data. Which is not a bad thing, necessarily. It is really important to understand these problems with simpler models before going to more complex models. However, I still do assert that "The main weakness of the paper is that it seems only tangentially related to the generative model collapse problem" as I stated in my original response. The author's rebuttal is effectively that others ([3],[4],[5]) have studied Gaussian data, and while that is true, those papers ALSO include experiments on non-Gaussian data. So, I personally think that the paper's results are weak for this case. Again, this does not mean that the paper is bad but rather that it has not been made clear how the **Gaussian results on regression** relate to **non-Gaussian data being generated**.
> >
> > I thank the authors again for addressing all of my other concerns.

---

> > > ### Author Response · Authors · 2024-08-10
> > >
> > > We thank the reviewer for the further discussion.
> > >
> > > - Being more specific about EXACTLY what is being demystified is important to make sure the work is well represented by the title.
> > >
> > > We would like to extend our general response paragraph that begins with, "Compared to prior works, our theoretical results offer a more detailed analysis." While we acknowledge the existence of theoretical worksinvolving Gaussian data, we argue that these studies have limitations and that the theoretical framework in this area can be significantly improved. Our work addresses it by developing an analytic theory that explicitly demonstrates how model collapse quantifiably emerges from training on synthetic data.
> > >
> > > For instance, [3] examines learning with **single-dimensional Gaussian** data, where the new distribution is generated by unbiased sample mean and variance estimators. However, model collapse is only analyzed via **lower bounds** rather than through analytic expressions. In contrast, our results provide an exact analytic characterization of model collapse in a more general setting. [5] conducts stability analysis at the distribution level with **infinite data**, focusing only on **local behavior**, which aligns more with fine-tuning rather than training from scratch. Moreover, **no actual learning is analyzed** in [5]. We, on the other hand, provide a detailed analysis of learning with finite samples and training from scratch.
> > >
> > > Similarly, [4] makes assumptions regarding how the synthetic distribution shrinks the generation distribution from $\Sigma$ to $\lambda\Sigma$. The **martingale property** of the unbiased estimators in their work leads to collapse almost surely as the number of generations approaches infinity. They do not analyze the collapse dynamics, such as its convergence rate or dependence on the number of samples. In contrast, our work offers a much more detailed and nuanced analysis that goes beyond simply establishing almost sure collapse.
> > >
> > > Furthermore, none of these three papers provide analysis on scaling laws, which are crucial to the success of foundation models. In contrast, our analysis in Section 4 offers precise insights into how scaling laws are affected by model collapse.
> > >
> > > In summary, prior works [3, 4, 5] lack the fine-grained analysis needed to fully understand model collapse, offering only the existence of the phenomenon through bounds in limited settings. Our work, in contrast, delivers a comprehensive analysis that reveals the underlying mechanisms of model collapse through exact formulae derived in the high-dimensional limit, across a wide range of settings. Notably, we are the first to demonstrate the breakdown of scaling laws, the impact of label noise, and the occurrence of model collapse even in noiseless settings. Our work provides a complete analytic picture of model collapse, justifying the use of the term "demystified".
> > >
> > > We will add all the above discussion.
> > >
> > > - It has not been made clear how the Gaussian results on regression relate to non-Gaussian data being generated.
> > >
> > > We would like to expand on our point that "numerous papers have already documented empirical observations of model collapse, and our findings are consistent with them." The math transformer experiment described in [2] aligns closely with our theoretical models. In this experiment, transformers were trained to predict the greatest common divisor (GCD) between two numbers, with predictions generated using next-token prediction. Importantly, the distribution in this case is clearly not Gaussian, yet the experimental results still correspond well with our theoretical predictions.
> > >
> > > Specifically, the middle plot of Figure 4 in [2] closely matches the right plot of Figure 1(a) in our paper. The gap between AI-generated data and real data in their plot is predicted by our Theorem 3.6 and the scaling behavior described in Theorem 4.1. Additionally, the left plot of Figure 4 in [2] reflects the breakdown of scaling laws that we predicted.
> > >
> > > We could potentially include a similar result in our paper after the rebuttal period and before the camera-ready version.
> > >
> > > Moreover, we would like to emphasize that the use of a Gaussian design in our setting primarily serves to obtain a precise characterization. However, there is no significant difference between Gaussian and non-Gaussian distributions that would hinder generalization of the general insights. Existing research indicates that high-dimensional features often exhibit a degree of Gaussianity [1], and Gaussian data with the same mean and variance can effectively predict performance—a concept known as **Universality** in the literature.

---

> > > > ### Author Response · Authors · 2024-08-10
> > > >
> > > > *Our contribution is intentionally theoretical, not experimental.*
> > > >
> > > > Indeed, this is a theory paper, in the sense that we propose to develop a effective theory which explains the emergence of the model collapse phenomenon. As such we focus on developing theory, providing valuable insights through comprehensive analyses across various settings modelled to capture relevant aspects of the real world, and supplemented by empirical validation. Given the existing empirical evidence in the literature, space constraints, and the current lack of a complete theoretical framework in this area, we have chosen to present our work with a primary focus on theory.
> > > >
> > > > [1] Hu, Hong, and Yue M. Lu. "Universality laws for high-dimensional learning with random features." IEEE Transactions on Information Theory 69.3 (2022): 1932-1964.
> > > >
> > > > [2] Elvis Dohmatob, et al. "A Tale of Tails: Model Collapse as a Change of Scaling Laws." ICML 2024.
> > > >
> > > > [3] Ilia Shumailov, et al. "AI models collapse when trained on recursively generated data." Nature 2024.
> > > >
> > > > [4] Sina Alemohammad, et al. "Self-Consuming Generative Models Go MAD." ICLR 2024.
> > > >
> > > > [5] Quentin Bertrand, et al. "On the stability of iterative retraining of generative models on their own data." ICLR 2024.

---

> > > > ### Comment · Reviewer_16H4 · 2024-08-12
> > > > **Response to rebuttal #2**
> > > >
> > > > Thanks again for the response.
> > > >
> > > > Regarding using the word "demystified": I acknowledge that this work addresses novel areas of the Gaussian setting. And the authors have done a good job doing so. I still maintain that the "demystified" part of the paper is ambigous. The rebuttal that the authors mentioned is essentially distinguishing the previous work to their and although that is fine, it doesn't really address the concern that word does not describe very well what the paper is actually about. Something like "Model Collapse Asymptotics in The Case of Gaussian Regression," for example, would be considerablely more descriptive of what the paper is about.
> > > >
> > > > Regarding how these results apply practically: The authors mention that their findings are consistent with model collapse in practical, complex models. However, the question is how these **new findings from this work** are consistent with model collapse in practical, complex models? I don't think that the authors mean to say that these test error formulations that they develop hold for multi-layer neural networks in regression settings as well, do they? If I am mistaken then please correct me and I will retract this concern immediately and raise my score significantly. If there is no barrier to go from non-Gaussian results to Gaussian due to the Gaussianity of features, I welcome the authors to do so.
> > > >
> > > > Thank you.

---

> ### Author Response · Authors · 2024-08-13
> **Response #1**
>
> We thank the reviewer for the further engagement.
>
> ### 1. Regarding using the word "demystified"
>
> We are happy to accept the proposed title to reduce the emphasis on "demystify."
>
> ### 2. Regarding how these results apply practically
> >*I don't think that the authors mean to say that these test error formulations that they develop hold for multi-layer neural networks in regression settings as well, do they?*
>
> Let us clarify. Regarding complex neural network models
>
> - Our theory (i.e linear increase in test error as a function of the number of iterations $n$) predicts their behavior in the linearized regimes (finite-width random features and NTK regimes, etc.). This is because such models are essentially kernel regression [1, 2]. Specifically for the RF regime, we empiriclally confirm this with additional results given in **Table 1** below, which complement the results of **Appendix C.2** of our manuscript (corresponding to classical RBF and polynomial kernels).
> - For neural networks with finite widths (and fully trained), our theory doesn't direcly apply. Our speculation is that such a collapse continues to occur (consistently with empirical observations in the literature, albeit for LLMs based on transformers). We anticipate that the general trends uncovered by our asymptotic theory will hold true—for example, more parameters is expected to lead to greater model collapse, as shown in **Theorem 3.1** and demonstrated in the following **Table 2**. More on this latter. Thus, ***model collapse in neural networks is even more severe than in linear regression.***
>
> However, it is quite possible that for general NNs (finite width, full-trained, etc.) amount of model collapse as a function of number of iterations $n$ might switch between linear increase (our current theory) to, quadratic, etc., depending on properties of the network like the the width, or other design choices.
>
> To explore this hypothesis further, we conducted a series of experiments over **the past 24 hours**. Our aim was to extend the analysis beyond linear settings and Gaussian data by employing a two-layer neural network with ReLU activation on the MNIST dataset. The models were trained using stochastic gradient descent (SGD) with a batch size of 128 and a learning rate of 0.1. We employed a regression setting where labels were converted to one-hot vectors, and the model was trained using mean squared error for 200 epochs to convergence. For each data synthetic generation, Gaussian label noise with a standard deviation of 0.1 is added. The test error is consistently evaluated on the test set using the clean labels.
>
> We considered two scenarios:
> - (1) learning with random features (RF) models, where the first layer was fixed randomly, and only the second layer was trained, and
> - (2) learning with a fully trainable neural network.
>
> The results for RF models of width (i.e number of hidden dimensions) $k$ of 20,000 are presented in the table below (figures cannot be uploaded). For clarity, an additional column showing the test error gap between different generations (indicated by $n$) is included.
>
> **Table 1**. Performance of very wide RF model on MNIST, with one-hidden layer NN (width $k=20,000$). Standard deviation is calculated using 10 random seeds.
>
> | Generation | Mean   | Std    | Diff   |
> |-------|--------|--------|--------|
> | 0     | 0.0136 | 0.0003 | 0.0050 |
> | 1     | 0.0186 | 0.0006 | 0.0024 |
> | 2     | 0.0210 | 0.0005 | 0.0021 |
> | 3     | 0.0232 | 0.0009 | 0.0018 |
> | 4     | 0.0250 | 0.0020 | 0.0012 |
> | 5     | 0.0262 | 0.0021 | 0.0020 |
> | 6     | 0.0281 | 0.0031 | 0.0012 |
> | 7     | 0.0293 | 0.0039 | 0.0021 |
> | 8     | 0.0314 | 0.0054 | 0.0022 |
> | 9     | 0.0335 | 0.0066 | -      |
>
> We observe that, with the exception of the first two generations, the decay in MSE loss generally follows a linear trend, which is consistent with the predictions of our theory.

---

> > ### Author Response · Authors · 2024-08-13
> > **Response #2**
> >
> > Next, we consider the scenario of training the full neural network. By varying the the width $k$, we change the number of parameters to further investigate the theoretical predictions. The following tables present the MSE loss for each generation ($n$) and the difference between generations (Diff).
> >
> > **Table 2**. The performance of two-layer neural network on MNIST with varying hidden dimensions.
> >
> > | Width (k) | n= 0 | n=1 | n=2 | n=3 | n=4 | n=5 | n=6 | n=7 | n=8 | n=9 | n=10 | n=11 |
> > |----------|--------|--------|--------|--------|--------|--------|--------|--------|--------|--------|--------|--------|
> > | 100      | 0.0082 | 0.0093 | 0.0103 | 0.0109 | 0.0128 | 0.0161 | 0.0209 | 0.0275 | 0.0357 | 0.0461 | 0.0571 | 0.0705 |
> > | 200      | 0.0071 | 0.0089 | 0.0102 | 0.0108 | 0.0145 | 0.0209 | 0.0302 | 0.0434 | 0.0619 | 0.0816 | 0.1123 | 0.1492 |
> > | 800      | 0.0065 | 0.0108 | 0.0137 | 0.0183 | 0.0294 | 0.0495 | 0.0783 | 0.1099 | 0.1634 | 0.2529 | 0.3573 | 0.4856 |
> > | 4000     | 0.0069 | 0.0134 | 0.0177 | 0.0223 | 0.0383 | 0.0661 | 0.1134 | 0.1745 | 0.2518 | 0.3429 | 0.4357 | 0.5893 |
> >
> > **Table 3**. The performance difference of the two-layer neural network  (of varying width $k$) between consecutive generations $n$.
> >
> > | Width (k) | Diff 0 | Diff 1 | Diff 2 | Diff 3 | Diff 4 | Diff 5 | Diff 6 | Diff 7 | Diff 8 | Diff 9 | Diff 10 |
> > |----------|--------|--------|--------|--------|--------|--------|--------|--------|--------|--------|--------|
> > | 100 | 0.0011|0.0010|0.0006|0.0019|0.0032|0.0047|0.0066|0.0082|0.0104|0.0109|0.0133
> > | 200 | 0.0017|0.0013|0.0006|0.0037|0.0063|0.0093|0.0132|0.0184|0.0197|0.0306|0.0369
> > | 800 | 0.0043|0.0029|0.0046|0.0110|0.0201|0.0287|0.0316|0.0534|0.0895|0.1043|0.1282
> > | 4000 | 0.0065|0.0043|0.0046|0.0160|0.0278|0.0472|0.0610|0.0773|0.0910|0.0928|0.1536
> >
> > **Observations.** From the table, we can observe that
> >
> > - More parameters (wider neural networks, i.e large $k$) lead to increased model collapse. This observation is consistent with our results proved linear regime (e.g Theorem 3.1). For linear models, the number of parameters is proportional to $d$ (the input dimension), whereas in two-layer neural networks, the "number of parameters"  is of order $kd$ (i.e proportional to the width $k$).
> >
> > - The dependence of model collapse on the number of iterations $n$ is linear for small values of $n$ (with $n \leq 4$ in our experiments), and becomes superlinear (possibly quadratic) for larger values of $n$ (with $n \geq 4$). Recall that $n=0$ corresponds to training on clean data from the data distribution. Thus, possibly, model collapse neural networks appears to be even more severe than in linear regression. A rigorously theoretically analysis of these new phenomena will be done in a subsequent work.
> >
> > We thank the reviewer for trigering the above discussion, which will be included in the manuscript.
> >
> > [1] Arthur Jacot, Franck Gabriel, and Clément Hongler. "Neural tangent kernel: Convergence and generalization in neural networks." Advances in neural information processing systems 31 (2018).
> >
> > [2] Sanjeev Arora, et al. "Fine-grained analysis of optimization and generalization for overparameterized two-layer neural networks." International Conference on Machine Learning. PMLR, 2019.

---

> > > ### Comment · Reviewer_16H4 · 2024-08-13
> > > **Response to rebuttal #3**
> > >
> > > Thank you for including these experiments! It is indeed very interesting that the neural network setting seems to behave differently than in the linear setting, namely worse. I do think that these experiments do provide evidence that the work on Gaussians is useful in the general setting, at least by being a lower bound on the kind of error once can get.
> > >
> > > And thank you for agreeing to change the title to be more descriptive.
> > >
> > > I will update my score now and I hope that your paper gets accepted!

---

> > > > ### Author Response · Authors · 2024-08-13
> > > >
> > > > Thank you for your appreciation and support! We will include all the new results in the final version of the paper.

---

### Author Rebuttal · Authors · 2024-08-07

We appreciate the time and effort the reviewers have dedicated to reviewing our paper. We are delighted to see that overall reception has been positive, with all reviewers acknowledging our theoretical contributions, particularly in analytically characterizing model collapse across a wide range of settings and our findings on scaling laws. Building on extensive work in the classical (non-iterative) case [1], we demonstrate how previously understood phenomena related to scaling and optimal regularization are altered with synthetic data. These insights, derived from comprehensive analyses using random matrix theory, align with empirical observations [2, 11], despite the simplicity of our model and the use of Gaussian data (with general covariance). The findings on scaling laws are particularly relevant to practitioners, as they are foundational for large language models. We hope this work serves as a catalyst for further studies in the field.

One common concern among reviewers is whether our results, based on Gaussian data and linear regressions, generalize to modern generative models for images and languages. We would like to clarify our focus on theoretical aspects. In the area of model collapse, there exists a substantial body of literature [3, 4, 5], that provides theoretical analyses using Gaussian data. For example, [6] examines the mixing effects of combining real and synthetic data in generative models using a linear model, Gaussian data, and asymptotic analysis. Additionally, [7] explores scaling laws in linear regression with Gaussian data, revealing phenomena that align with empirical observations. Thus, the use of Gaussian data and linear regression is a standard approach that can offer valuable insights even for large models.

Moreover, regression is not limited to traditional applications; current large language models (LLMs) are also utilized in labeling tasks. Many tasks, such as question-answering (QA), code generation, and mathematical problem-solving, involve inputs in the form of questions x and responses y. This provides a rationale for extending our results to real-world applications, as we have observed strong alignment with findings from other empirical studies. Furthermore, there is ample evidence [8, 9, 10] suggesting that LLMs exhibit linear behavior in their handling of concepts. Thus, regression with complex kernel functions serves as a good theoretical model.

Our work specifically focuses on high-dimensional linear regression on multivariate data because it allows us to: (1) achieve a solvable setting, and (2) explore a sufficiently rich model (through the covariance matrix $\Sigma$ and the relative scaling of sample size $n$ and and dimensionality $d$). This approach allows us to abstract away all the nitty gritty of neural networks and large language models, while enabling us to present a clear analytic picture which explains previously reported empirical findings on model collapse and provides an effective theory for this interesting phenomenon. Also, we outline fundamental limitations of synthetic data, the impact of label noise, and the degrees of over-/under-parameterization, etc.

Compared to prior works, our theoretical results offer a more detailed analysis. Prior studies have shown that model collapse occurs, but often without fine-grained analysis or only within a narrow range of settings. For instance, [3] shows model collapse with increasing variance, which approximates our results in an over-parameterized setting. [5] provides stability analysis at the distribution level with infinite data, focusing only locally, aligning with fine-tuning rather than training from scratch. We appreciate reviewer 16H4's observation that [4] relies on martingale techniques at the expectation level. In contrast, our work applies to general cases, providing analytic results to fully understand model collapse. We demonstrate how scaling laws change as a consequence, how optimal regularization shifts, and that model collapse can occur even without noise, expanding beyond the theories presented in [3]. These insights provide a clear understanding that was not achievable with previous approaches.

Some reviewers also suggested adding more empirical experiments. As discussed, numerous papers have already documented empirical observations of model collapse. Our primary contribution lies in the theoretical domain, where we are the first to fully characterize model collapse to this extent. Our goal is to comprehensively understand the factors contributing to model collapse and their effects on scaling laws and regularization. Given the depth of our theoretical analysis, we believe additional empirical results would be superfluous. Our findings on model collapse and scaling laws are consistent with the experiments reported in [7, 8], which explored generative models in both mathematical and natural language contexts, and in [4, 5], which investigated generative models for images.

In conclusion, we are grateful for the constructive feedback provided by the reviewers. Our work not only addresses significant theoretical questions but also aligns closely with empirical findings, providing a general framework for understanding complex phenomena in model collapse.

[1] Cui, Hugo, et al. "Generalization error rates in kernel regression: The crossover from the noiseless to noisy regime." NIPS 2021.

[2] Elvis Dohmatob, et al. "A Tale of Tails: Model Collapse as a Change of Scaling Laws." ICML 2024.

[3] Ilia Shumailov, et al. "AI models collapse when trained on recursively generated data." Nature 2024.

[4] Sina Alemohammad, et al. "Self-Consuming Generative Models Go MAD." ICLR 2024.

[5] Quentin Bertrand, et al. "On the stability of iterative retraining of generative models on their own data." ICLR 2024.

[6] Ayush Jain, et al. "Scaling laws for learning with real and surrogate data."

[7] Licong Lin, et al. "Scaling Laws in Linear Regression: Compute, Parameters, and Data."

---

> ### Author Response · Authors · 2024-08-07
>
> [8] Neel Nanda, et al. "Emergent Linear Representations in World Models of Self-Supervised Sequence Models." EMNLP 2023.
>
> [9] Kiho Park, et al. "The Linear Representation Hypothesis and the Geometry of Large Language Models." ICML 2024.
>
> [10] Yibo Jiang, et al. "On the Origins of Linear Representations in Large Language Models." ICML 2024.
>
> [11] Yunzhen Feng, et al. "Beyond Model Collapse: Scaling Up with Synthesized Data Requires Reinforcement."

---

### Decision · Program_Chairs · 2024-09-25

**Decision:**

Accept (poster)

**Comment:**

This paper provides a theory for the model collapse phenomenon in the high dimensional regime of linear regression and Gaussian data. In addition, the paper examines the mitigation of model collapse by adjustable regularization. Although the examined regression setting has conceptual differences from model collapse in generative models, the contributions are of sufficient interest and significance and can be considered for publication.

During the rebuttal and Authors-Reviewers discussion, the authors significantly improved their paper following detailed comments and discussion with Reviewers 16H4 and jLLV. This improvement includes better description of details and claims regrading the contributions in this paper. Moreover, the original submission did not consider the connection of its linear model theory to nonlinear models. In the rebuttal, following comments by Reviewer 16H4, the authors added experiment results showing that their theoretical principles appear (to some extent) in nonlinear two-layer neural networks. By this, the authors have adequately addressed this concern. The authors are requested to update their paper according to their discussions with the reviewers.

Additional requests for the authors to implement in their camera-ready version of the paper:
1. The authors claim in the paper that their results for linear regression can be *straightforwardly* extended to kernel regression. Reviewer jLLV commented about this and discussed this with the authors.  At the end of this discussion, the authors wrote that `` We agree with the reviewer that the statement about kernel regression should be a remark that our analysis can be straightforwardly extended to the case of kernel regression. This will be rectified.’’ The authors mentioned that they discuss the extension to kernel regression in Appendix B --- There they explain that it is *straightforward* to extend their results to kernels by applying the Gaussian assumption on the feature vector instead of the input vector. But this seems too simplistic, because the extension should still consider a Gaussian input and then for a nonlinear feature map the feature vector will not be Gaussian. Their assumption that the feature vector is Gaussian can hold in overly simple cases, for example, if the input is Gaussian and the feature map is linear. Alternatively, it might be that this assumption could be justified by another, more complex explanation than what currently available in the paper. Accordingly, the authors are requested to remove from the camera-ready version the claim that their results are *straightforwardly* extended to kernel regression, as well as the explanation in Appendix B. This request is also supported by further discussion I had with reviewers. Importantly, the removal of this claim from the paper does not reduces its contribution and significance.
2. The authors are requested to further search for typos and fix them. This includes the appendices and the proofs. For example, please check the proof in Appendix E.2 (the equations after line 625), as there might be a typo in the subscript numbering in the equality between the last expression of the first line of equations to the first expression in the second line of equations.

Based on reading the paper, the reviews, the authors’ rebuttal and their discussion with the reviewers, I find this paper of sufficient quality and significance for the theory of contemporary machine learning. My recommendation is therefore to accept this paper.